# Crosslinking-guided geometry of a complete CXC receptor-chemokine complex and the basis of chemokine subfamily selectivity

**Tony Ngo** [ORCID], **Bryan S. Stephens, Martin Gustavsson** [ORCID], **Lauren G. Holden** [ORCID], **Ruben Abagyan, Tracy M. Handel** [ORCID]\*, **Irina Kufareva** [ORCID]\*

Skaggs School of Pharmacy and Pharmaceutical Sciences, University of California San Diego, La Jolla, California, United States of America

\* thandel@health.ucsd.edu (TMH); ikufareva@health.ucsd.edu (IK)

⊙ OPEN ACCESS

**Data Availability Statement:** All relevant data, including the coordinates of the CXCR4-CXCL12

## Abstract

Chemokines and their receptors are orchestrators of cell migration in humans. Because dysregulation of the receptor-chemokine system leads to inflammation and cancer, both chemokines and receptors are highly sought therapeutic targets. Yet one of the barriers for their therapeutic targeting is the limited understanding of the structural principles behind receptor-chemokine recognition and selectivity. The existing structures do not include CXC subfamily complexes and lack information about the receptor distal N-termini, despite the importance of the latter in signaling, regulation, and bias. Here, we report the discovery of the geometry of the complex between full-length CXCR4, a prototypical CXC receptor and driver of cancer metastasis, and its endogenous ligand CXCL12. By comprehensive disulfide cross-linking, we establish the existence and the structure of a novel interface between the CXCR4 distal N-terminus and CXCL12 β1-strand, while also recapitulating earlier findings from nuclear magnetic resonance, modeling and crystallography of homologous receptors. A cross-linking–informed high-resolution model of the CXCR4-CXCL12 complex pinpoints the interaction determinants and reveals the occupancy of the receptor major subpocket by the CXCL12 proximal N terminus. This newly found positioning of the chemokine proximal N-terminus provides a structural explanation of CXC receptor-chemokine selectivity against other subfamilies. Our findings challenge the traditional two-site understanding of receptor-chemokine recognition, suggest the possibility of new affinity and signaling determinants, and fill a critical void on the structural map of an important class of therapeutic targets. These results will aid the rational design of selective chemokine-receptor targeting small molecules and biologics with novel pharmacology.

## Introduction

As orchestrators of cell migration, chemokines and their receptors are critical to many physiological and disease-related processes, including embryogenesis and organ development, immune surveillance, inflammation, and cancer metastasis [1]. Chemokines are secreted 8 to

complex presented here, are within the paper and its Supporting Information files.

**Funding:** National Institute of Allergy and Infectious Diseases, https://www.niaid.nih.gov/: R01 AI118985 to I.K. and T.M.H. National Institute of General Medical Sciences, https://www.nigms.nih.gov/: R01 GM117424 to I.K. and T.M.H. National Health and Medical Research Council, https://www.nhmrc.gov.au/: C.J. Martin Early Career Fellowship 1145746 to T.N. National Institute of General Medical Sciences, https://www.nigms.nih.gov/: T32 GM007752 to B.S.S. National Institute of General Medical Sciences, https://www.nigms.nih.gov/: T32 GM008326 to B.S.S. Robertson Foundation/Cancer Research Institute Irvington https://www.cancerresearch.org/: a postdoctoral fellowship to M.G. National Heart, Lung, and Blood Institute, https://www.nhlbi.nih.gov/: R35 HL135737 to I.K. National Institute of Neurological Disorders and Stroke, https://www.ninds.nih.gov/: R01 NS102432 to I.K. National Institute of General Medical Sciences, https://www.nigms.nih.gov/: R01 GM071872 to R.A. The funders had no role in study design, data collection and analysis, decision to publish, or preparation of the manuscript.

**Competing interests:** The authors have read the journal's policy on competing interests. R.A. has an equity interest in Molsoft, LLC. The terms of this arrangement have been reviewed and approved by the University of California San Diego in accordance with its conflict of interest policies. All other authors declare that no competing interests exist.

**Abbreviations:** BPMC, biased probability Monte Carlo; BRET, bioluminescence resonance energy transfer; CHS, cholesteryl hemisuccinate; CRS, chemokine recognition site; DDM, n-dodecyl-ß-D-maltopyranoside; GMFI, geometric mean fluorescence intensity; GPCR, G protein-coupled receptor; HEK293T, human embryonic kidney 293T cells; IU, infectious units; NMR, nuclear magnetic resonance; Rluc3, Renilla luciferase 3; SASA, solvent-accessible surface area; Sf9, *Spodoptera frugiperda*; TM, transmembrane; WT, wild type.

12 kDa proteins that share a conserved topology and are classified into 4 subfamilies (CC, CXC, $CX_3C$, and XC) based on the number and arrangement of cysteine residues in their N termini. Chemokine receptors are class A G protein-coupled receptors (GPCRs) that preferentially recognize chemokines from individual subfamilies. However, within their subfamily, many receptors respond to multiple chemokines, often in different or biased ways [2]; additionally, there are examples of receptor-chemokine interactions across subfamilies, notably involving atypical and virally encoded receptors and chemokines [3]. The complexity of the resulting interaction and signaling network has been cited as a reason for the failures of numerous chemokine receptor-targeting clinical candidates, with only 2 small molecule drugs (CXCR4 and CCR5 antagonists) approved by the FDA thus far [2,4,5]. Another reason is the limited understanding of structural principles that determine receptor-chemokine recognition and selectivity, which poses challenges for the development of anti-chemokine therapeutics with desired pharmacological profiles.

Recently, X-ray structures of several receptor complexes with CC and $CX_3C$ chemokines were determined and have provided insight into general principles of chemokine binding and receptor activation [6–9]. The structures revealed extensive and compositionally complex interaction interfaces with functionally distinct parts that confirm and expand the canonical "two-site" hypothesis of receptor-chemokine recognition [10–14]. The two classical parts of the interface are chemokine recognition site 1 (CRS1) where the proximal N terminus of the receptor binds the globular core of the chemokine and CRS2 where the transmembrane (TM) domain binding pocket of the receptor interacts with the N-terminal signaling domain of the chemokine. The structures also revealed an unexpected intermediate anchoring site (CRS1.5) between the conserved disulfide bond of the chemokine and the conserved proline-cysteine motif in the proximal N-terminus of the receptor.

Despite these advances, our understanding of chemokine-receptor interactions remains incomplete for two reasons. First, CXC chemokine complexes have not yet yielded to structure determination efforts. Also, the distal N-termini of receptors are invariably disordered in crystal structures solved thus far. For many receptor-chemokine complexes, the N-terminus is known to play an important role in affinity and selectivity [15–18], both of which may be further fine-tuned by posttranslational sulfation of tyrosine residues [19,20]. Furthermore, the distal N-terminus has been proposed to be a determinant of chemokine-mediated signaling [18,21,22] and to contribute to signaling bias [23]. These studies established the functional importance of receptor N-termini; however, their interaction geometry with chemokines remains elusive (S1 Table), which is a critical barrier for understanding how chemokines control diverse signaling events. In complementary studies, the interaction geometries have been explored by nuclear magnetic resonance (NMR) with isolated N-terminal peptides of receptors [24–31] (S1 Table). Although these efforts have captured some important residue pairings, the absence of spatial constraints from the TM domain of intact receptors has led to geometric inconsistencies across the NMR structures and in relation to crystallographic structures of the TM domains [32,33].

Here, we aimed to resolve these inconsistencies and characterize the geometry of the complex of full-length CXC chemokine receptor, CXCR4, and its endogenous chemokine CXCL12, via comprehensive disulfide cross-linking (S1 Table) and cross-linking-guided molecular modeling. Building upon prior experimental and modeling efforts that focused on CRS2 [6,33], we refined the model and completed it by revealing the structural basis for the binding of the full receptor N-terminus to various regions of the chemokine globular core. Consistent with pharmacological evidence of the involvement of the receptor distal N-terminus in chemokine recognition [18,22], we elucidated the geometry of its interactions with the chemokine β1-strand and validated the so-called CRS0.5 previously proposed for a

homologous receptor [34]. The complete model provides insights into the role of select polar residues in the receptor and the chemokine and explains the effects of receptor tyrosine sulfation. Furthermore, as complexes containing CXC chemokines have not yet been crystallized, the present experimentally validated full-length CXC receptor-chemokine model is valuable for the analyses of recognition specificity across the major chemokine families.

## Results

### A structural hypothesis of the CXCR4-CXCL12 interaction guided by CXCL12 chemical properties

Disulfide crosslinking [6,35], our chosen approach to reveal the geometry of the CXCR4-CXCL12 complex, involves co-expression of pairs of receptor and chemokine Cys mutants in *Spodoptera frugiperda (Sf9)* insect cells and monitoring the formation of covalent complexes (S1 Fig). Because comprehensive all-to-all pairwise mutagenesis (estimated 54 chemokine core residues × 25 receptor N-terminus residues = 1,350 pairs) is impractical, we sought guidance for the placement of the CXCR4 N-terminus from structural and chemical properties of CXCL12 itself. Peptide binding sites on globular proteins often combine a rigid and hydrophobic center with a flexible polar rim [36,37] or utilize β-sheet-like backbone interactions [38,39]. Accordingly, we evaluated these properties for the exposed structural elements of the CXCL12 globular core (the N-loop, $3_{10}$ helix, β1-strand, 30s loop, 40s loop, and β3-strand; Fig 1A and 1B). Mapping was performed using multiple crystal structures of CXCL12 and yielded the following observations (Fig 1A–1I; S2 Table):

- High degree of solvent exposure and flexibility of residues in the N-loop (12-RFFESH-17), particularly H17, suggest likely involvement in protein-protein interactions (Fig 1A and 1D–1G). Such exposure and dynamics are also consistent with fast radiolytic oxidation of these residues in an earlier CXCL12 footprinting experiment [34].

- High degree of solvent exposure and flexibility also characterize the 40s loop, including the asparagine cluster 44-NNN-46 (Fig 1A and 1D–1G).

- The N-loop and the 40s loop together frame a groove (Fig 1D, 1F and 1H, black stroke) on the chemokine surface that has a base lined by the rigid and almost completely buried aliphatic residues from the β3-strand; this composition (rigid hydrophobic bottom and polar flexible rims) is common in peptide-binding interfaces [36,37].

- The β1-strand (25-HLKIL-29) presents a "path" of solvent-accessible residue backbones (Fig 1I, black stroke), whereas the side chains have limited solvent accessibility due to secondary structure constraints (Fig 1A and 1E); this arrangement favors backbone-backbone interactions with potential binding partners. In fact, the β1-strand of CXC chemokines mediates homodimerization [40] and may be "repurposed" for binding to the receptor (supported by modeling [41] and the recent NMR structure with the N-terminal peptide of CXCR4 [25]) or other proteins [42].

- The mostly buried aliphatic residues L26 and I28 in the β1-strand (Fig 1A and 1E) likely have a structural role in supporting the integrity of the β-sheet.

In the available CXCL12 crystal structures, the N-loop, 30s loop, and 40s loop adopt distinct conformations in a coordinated fashion (Fig 1A and 1C; S3 Table). It is not known whether all or only some of the resulting principal conformations of CXCL12 are relevant for its

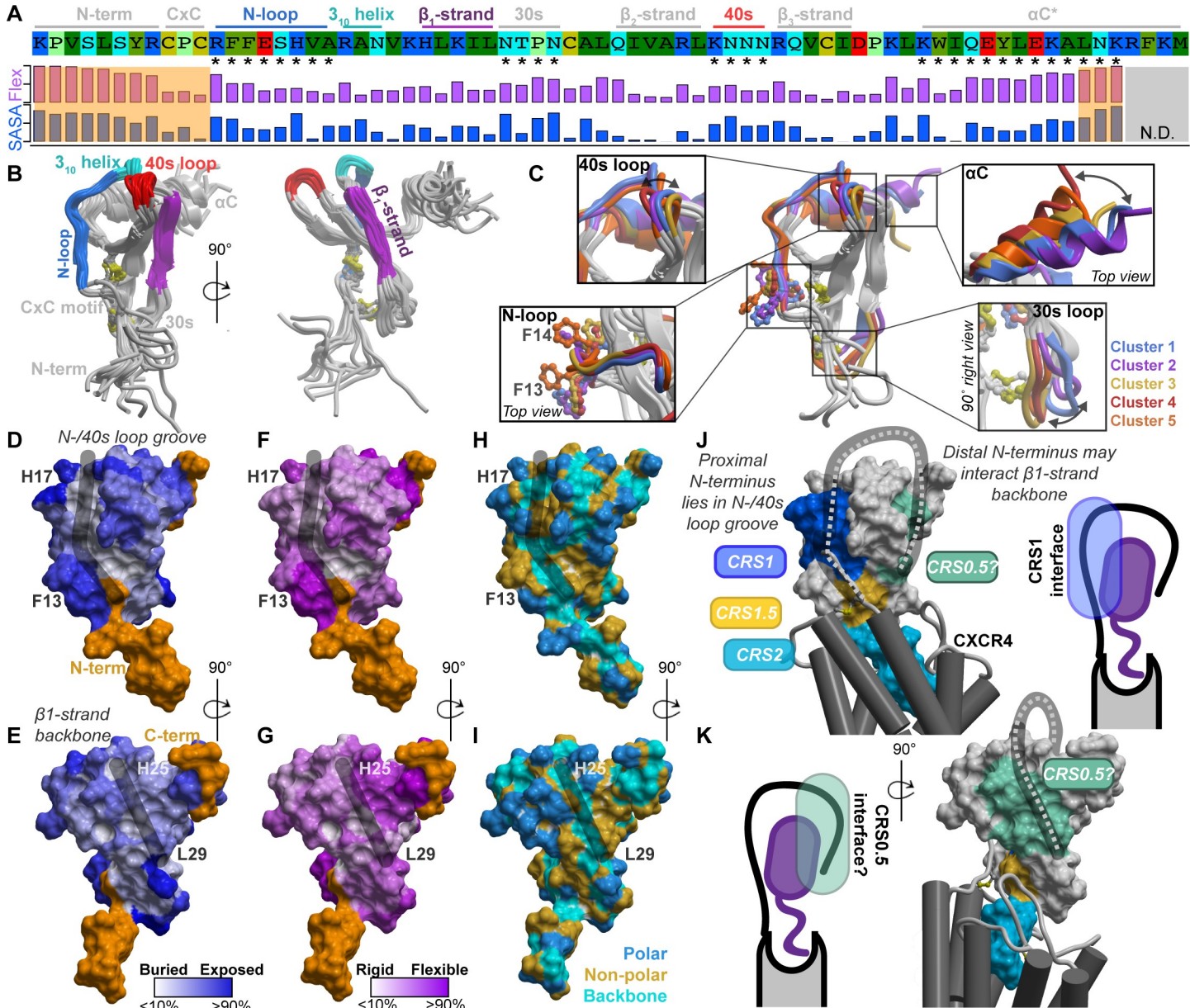

**Fig 1. The structural and physicochemical properties of the chemokine CXCL12.** (A) The sequence of CXCL12, its annotated structural domains, and quantification of side chain conformational variability and normalized SASA. The values are given in S2 Table. (B) The exposed structural elements of CXCL12 highlighted on the crystallographic ensemble of structures. (C) Coordinated conformations of CXCL12 structural elements, which are indicated by asterisks in panel A, captured by clustering of the 30s loop. (D–G) Normalized residue side chain SASA (D–E) and side chain conformational variability (F–G) mapped onto the CXCL12 surface. The highly flexible N- and C-termini are colored orange. (H–I) Exposed residue backbones, polar and nonpolar side chains on the CXCL12 surface. In panels D–I, the black strokes highlight surface grooves in CXCL12. (J–K) A coarse-grain hypothesis of CXCR4 (gray) bound to CXCL12 (surface mesh is colored by known and proposed CRS between the receptor and chemokine). The hypothesis is based on published CRS2 models and the presented structural and physicochemical properties of CXCL12. CRS, chemokine recognition site; SASA, solvent-accessible surface area.

interaction with the receptor, which may have implications for generating a high-resolution model of the complex (addressed below).

Altogether, this analysis identified 2 patches on the chemokine surface whose properties made them favorable for interactions with the receptor N-terminus: the N-loop/40s loop groove on one side and the β1-strand on the other. Taking into account the structural

constraints imposed by the previously published partial complex models [6,43], we put forward a low-resolution hypothesis to broadly delineate the interaction interfaces and guide selection of residue groups for cross-linking. According to this hypothesis, the proximal N-terminus of CXCR4 (CRS1) binds in the chemokine's N-loop/40s loop groove (Fig 1J and 1K, dark blue), while the rest of the N-terminus wraps around the chemokine to interact with the CXCL12 β1-strand, potentially extending the chemokine β-sheet (Fig 1J and 1K, light green). The latter interface is consistent with our earlier model of the homologous receptor, ACKR3 [34]. Given the abundance of positive charges on CXCL12 (Fig 1A and S2 Fig), we also postulated that the receptor-chemokine interaction is largely driven by electrostatics. Next, we sought to validate and refine this structural hypothesis using geometric constraints from disulfide cross-linking.

### Flow cytometry-based disulfide cross-linking identifies prominent CXCR4-CXCL12 residue proximities along the predicted interaction sites

First, the proposed interaction between the distal N-terminus of CXCR4 (residues 3-GISIY-7 and S9) and the β1-strand of CXCL12 (residues 25-HLKIL-29; Fig 1K) was systematically and comprehensively probed by disulfide cross-linking. To avoid any bias toward the initial hypothesis, we co-expressed every pair of receptor and chemokine cysteine mutants in this interface in *Sf9* cells (30 pairs total), after which covalent complexes were detected on the cell surface by flow cytometry via the chemokine C-terminal HA tag (S1 Fig). This was possible because, unless it is bound covalently, CXCL12 quickly dissociates from CXCR4 [44] and becomes undetectable on the insect cell surface [35]. In all cases, the antagonist variant [P2G] CXCL12 was used [10], because like many agonists, WT CXCL12 does not form high affinity complexes with the receptor, except in the presence of G protein [45–47]. In the cross-linking experiments, the previously crystallized disulfide cross-linked complex of CXCR4(D187C) with the viral chemokine antagonist vMIP-II(W5C) [6] was used as a positive control, and combinations of these mutants with other chemokine and receptor mutants as negative controls (12 additional mutant pairs).

The data demonstrated positive cross-links between [P2G]CXCL12 H25C, K27C, and L29C with the distal N-terminus of the receptor, confirming their proximity and interaction (Fig 2A and 2B). [P2G]CXCL12(L29C) strongly cross-linked with CXCR4(G3C) (107.2% efficiency compared to CXCR4(D187C)-vMIP-II(W5C); Fig 2A; S4 Table). Likewise, the G3C-K27C (135.5%), G3C-H25C (81.7%), Y7C-H25C (88.0%), and S9C-H25C (81.2%) receptor-chemokine combinations cross-linked comparably to CXCR4(D187C)-vMIP-II(W5C), whereas I4C-H25C, S5C-K27C, and S5C-H25C cross-linked moderately well (>50% efficiency; Fig 2A; S4 Table). Although the crosslinking pattern did not reveal a one-to-one correspondence between the receptor and chemokine residues at this interaction interface, definitive asymmetry was observed: CXCR4 S5C cross-linked with [P2G]CXCL12 K27C and H25C but not L29C (Fig 2A and 2B), whereas the only mutant to cross-link with L29C was CXCR4 G3C (Fig 2A and 2B). The observed crosslinking pattern supports an antiparallel geometry, in contrast to a parallel one, for the potential β-sheet between the CXCR4 N-terminus and CXCL12 β1-strand (Figs 1K and 2B).

Interestingly, all interactions involving [P2G]CXCL12 L26C and I28C mutants were strongly disfavored (Fig 2A). In control experiments with ACKR3 that not only binds [P2G] CXCL12 but also displays it on the *Sf9* cell surface without cross-linking because of slow off-rates [18,35], negligible surface chemokine was detected for [P2G]CXCL12 L26C and I28C (S3 Fig). This suggests that these mutant chemokines were misfolded, in agreement with the proposed structural role of L26 and I28 (Fig 1).

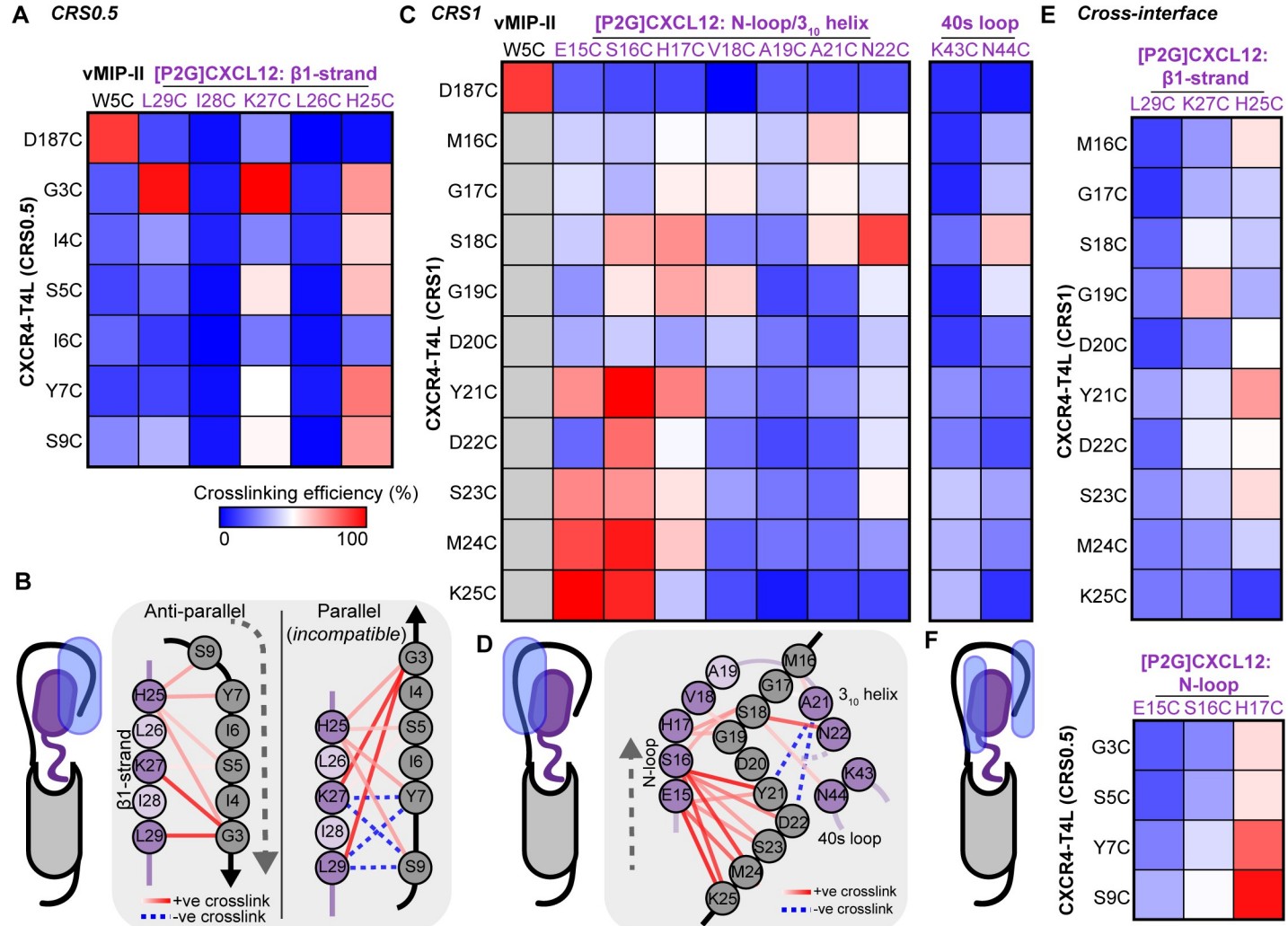

**Fig 2. CXCR4-[P2G]CXCL12 intermolecular crosslinking efficiencies detected by flow cytometry.** (A) A heat map of pairwise cross-linking efficiencies in the predicted CRS0.5, normalized to CXCR4(D187C)-vMIP-II(W5C), a previously crystallized covalent complex. (B) CRS0.5 crosslinking efficiencies mapped onto a 2D schematic of CRS0.5 in the hypothetical antiparallel and parallel β-sheet orientations. The data are more consistent with the antiparallel geometry. (C) A heat map of pairwise cross-linking efficiencies in CRS1, normalized to CXCR4(D187C)-vMIP-II(W5C). (D) CRS1 crosslinking efficiencies mapped onto a 2D schematic of CRS1. (E–F) Heat maps of pairwise cross-linking efficiencies when (E) Cys mutants in CXCR4 CRS1 are co-expressed with Cys mutants in CXCL12 β1-strand (CRS0.5) and (F) vice versa. Gray squares in the heat maps represent residue pairs that were not studied. Data represent the average of $n \geq 3$ independent biological replicates. The data in numerical form are provided in S4 Table, and the underlying numerical data for S4 Table and figure panels are in S1 Data. CRS, chemokine recognition site.

Next, we systematically probed the cross-linking propensities between CXCR4 proximal N-terminus residues (16-MGVGDYDSMK-25) and parts of the [P2G]CXCL12 N-loop and $3_{10}$ helix (15-ESHVARAN-22) (Fig 1J). Residue R20 was omitted from mutagenesis, because it is oriented in the opposite direction from the N-loop/40s loop groove in all CXCL12 structures and thus is unlikely to interact with the CXCR4 proximal N-terminus. We also generated CXCL12 40s loop mutants K43C and N44C and included the positive and negative controls as above. Again, to avoid bias, each-to-each receptor-chemokine residue pair cross-linking was attempted within this broadly defined patch, resulting in a total of 100 pairs. This cross-linking effort captured CXCR4 K25C cross-linking with [P2G]CXCL12 E15C and S16C, recapitulating previous observations with WT CXCL12 [33]. In fact, these residue pairs cross-linked with greater efficiency than the CXCR4(D187C)-vMIP-II(W5C) control (Fig 2C and 2D; S4 Table).

Residues Y21, S23, and M24 also cross-linked with [P2G]CXCL12 E15 and S16. Despite the cross-linking promiscuity of CXCR4 S18, it was the only residue to strongly cross-link with N22 and to moderately cross-link with N44 in the 40s loop. Furthermore, there was considerable (although less efficient) cross-linking between CXCR4 residues M16-S18 and [P2G] CXCL12 A21. These results provide strong evidence for the interaction between the proximal N-terminus of CXCR4 and the N-loop/40s loop groove of CXCL12 (Figs 1J, 2C and 2D).

To build confidence in the specificity of the flow cytometry findings and complete the cross-linking set, we co-expressed receptor and chemokine mutants across the predicted interaction sites, that is, CXCR4 CRS1 mutants with proposed [P2G]CXCL12 CRS0.5 mutants and vice versa. As expected, weak or negligible cross-linking was observed between the proximal N-terminus of CXCR4 (CRS1) and the CXCL12 β1-strand (Fig 2E). Similar to cross-linking in the proposed CRS0.5 interface (Fig 2A and 2B), H25 demonstrated a pattern of multiple weak cross-links, suggesting lack of specificity for this residue. An additional moderate cross-link was detected between CXCR4 G19C and [P2G]CXCL12 K27C, which was hard to reconcile in our hypothesized model; therefore, this pair was included in follow-up pull-down/Western blotting studies (below).

Unexpectedly, when probing the opposite combination, we observed considerable cross-linking between the distal N-terminus of CXCR4 (CRS0.5) and [P2G]CXCL12 H17C (Fig 2F). Similar to H25C, the lack of specificity for CXCL12 H17C may in part be explained by its unusually high solvent accessibility in the CXCL12 structures (Fig 1A and 1D; S2 Table). This proximity contradicts our initial structural hypothesis and suggests a potential alternative or a transient intermediate conformation in which the distal N-terminus of CXCR4 binds to the N-loop/40s loop groove of CXCL12; this conformation is likely facilitated by the similarity of the physicochemical properties in the distal and proximal N-terminal sequences of CXCR4 (2-EGISIYTS-9 versus 15-EMGSGDYDS-23). The strongest cross-interface pair of CXCR4 (S9C) with CXCL12(H17C) was also included in the follow-up pull-down experiments (below).

## Low throughput follow-up characterization of complexes by pull-down and Western blotting corroborates proximities identified by flow cytometry

Based on the flow cytometry data, a small number of receptor-chemokine combinations were selected for verification using an orthogonal and established method [6,34,35]. The cross-linked complexes were extracted from detergent-solubilized *Sf9* membranes via metal affinity pull-down using a His-tag on the receptor; the pulled-down complexes were separated on a nonreducing SDS-PAGE and probed for co-migrating chemokine by Western blotting (S1 Fig). These experiments included several pairs that appeared strongly cross-linked in the flow cytometry experiments (Fig 2), as well as select weakly cross-linked (Y21C-N22C and Y21C-N44C, Fig 2C) or non-cross-linked (K25C-A21C) combinations.

Consistent with expectations, the control pair of CXCR4(D187C) and vMIP-II(W5C) provided one of the brightest bands on anti-HA blots, indicating a large amount of HA-tagged chemokine co-migrating, and hence cross-linked, with the receptor (Fig 3A). Comparable amounts of co-migrating chemokine were observed when CXCR4(K25C) was co-expressed with [P2G]CXCL12 E15C and S16C, CXCR4(M24C) with [P2G]CXCL12(S16C) or CXCR4 M16C and G17C with [P2G]CXCL12(A21C) (Fig 3A and 3B): these pairs recapitulated flow cytometry findings in CRS1. The CRS1 pairs of CXCR4(Y21C) with [P2G]CXCL12 N22C and N44C and CXCR4(K25C) with [P2G]CXCL12(A21C), which had low anti-HA detection by flow cytometry, showed low-to-negligible co-migrating chemokine in this experiment as well (Fig 3A and 3B). For the CRS0.5 interface, the pull-down/Western blotting results reiterated

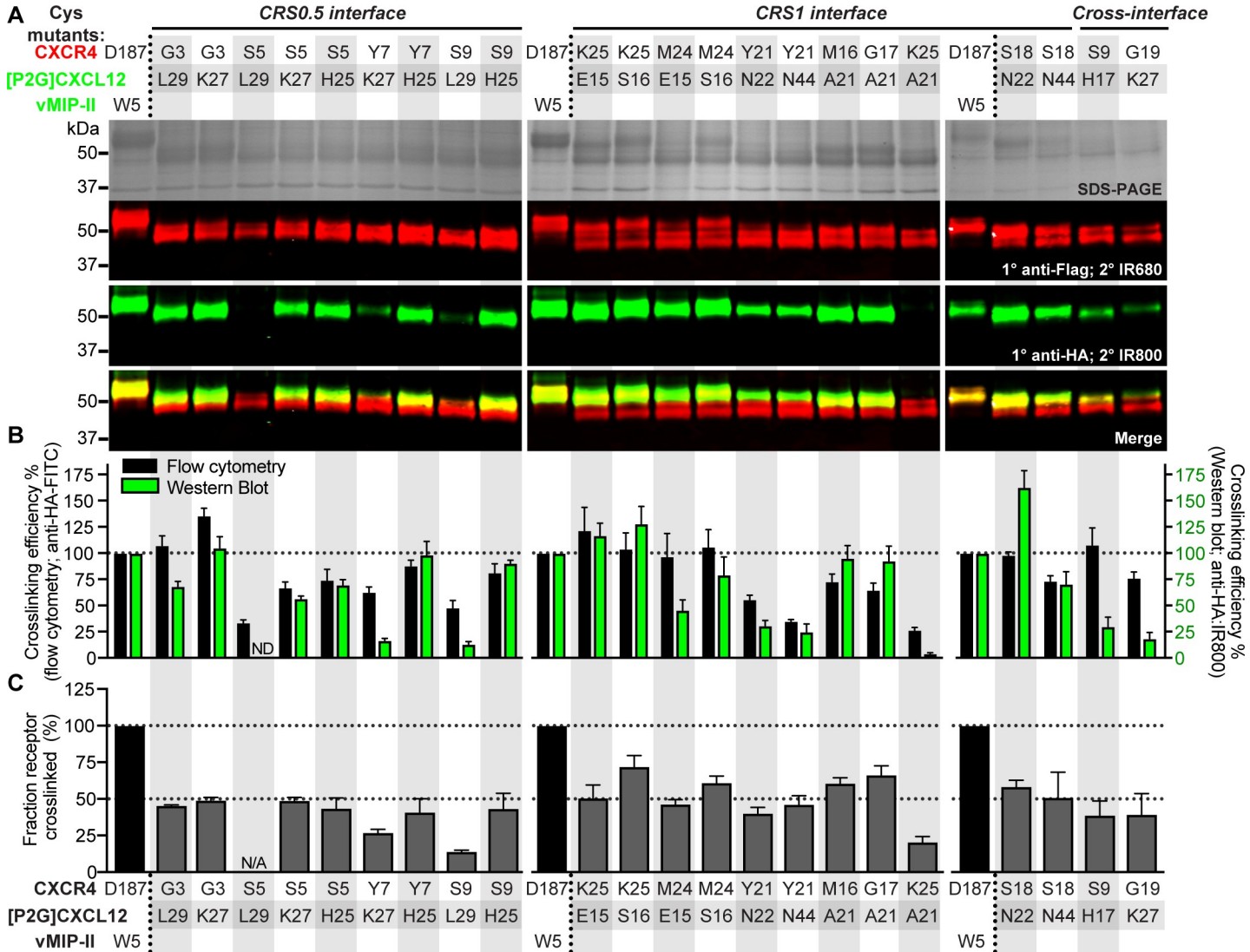

**Fig 3. CXCR4-[P2G]CXCL12 intermolecular cross-linking efficiencies detected by nonreducing SDS-PAGE and Western blotting of pulled down complexes.** (A) Nonreducing SDS-PAGE and Western blot of His-pull-downs of select combinations of CXCR4 and [P2G]CXCL12 cysteine mutants. The Flag-CXCR4-T4L receptor and [P2G]CXCL12-HA chemokine were detected by LI-COR IRDye conjugated secondary antibodies on a single blot (emission wavelength of 680 nm and 800 nm, visualized in red and green, respectively). Emitted fluorescence detected at 800 nm and 680 nm from the same band of the Western blot is indicative of the receptor and chemokine co-migrating on the gel and thus cross-linked. Images are representative of *n* = 3 independent replicates. (B) Comparison of cross-linking efficiency determined by flow cytometry (black bars) and Western blotting (green bars). Crosslinking efficiency by Western blotting is given by the anti-HA:IR800 fluorescence intensity. (C) The percentage of cross-linked receptor obtained from Western blotting of the pull-down samples, calculated as the ratio of the IR680 (red) signal intensity of the upper Flag band to the total receptor IR680 signal. Data in (B) and (C) are the mean and SEM of *n* = 3 independent replicates. The underlying numerical data for each figure panel can be found in S1 Data.

the antiparallel geometry of the CXCR4 N-terminus and [P2G]CXCL12 β1-strand (Fig 3A and 3B). In particular, the detection of the S9C-L29C complex, consistent with the alternative, parallel β-sheet geometry at this site (Fig 2B), was negligible (Fig 3A and 3B), suggesting that such geometry is unlikely or unstable. In general, cross-linked chemokine quantification by pull-down and Western blotting agreed well with that by flow cytometry, as indicated by their positive relationship (S4A Fig). This confirmed that flow cytometry, which is much less labor intensive, is sufficiently sensitive and predictive of the quantity of cross-linked complex in pulled-down samples. For future studies, flow cytometry detection alone should be sufficient

to map intermolecular residue proximities and to inform efforts of deducing receptor-chemokine interaction geometry by molecular modeling (below).

In addition to the amount of co-migrating chemokine, Western blotting allowed evaluation of an orthogonal property of the cross-linked complexes, namely, fraction of extracted receptor that was cross-linked with the chemokine, by comparing the intensities of receptor double bands on the blot. Such quantitation is not possible with flow cytometry. The control pair of CXCR4(D187C) and vMIP-II(W5C) formed nearly 100% cross-linked complex, as shown previously [6], whereas for other crosslinking pairs, distinct chemokine-bound and unbound receptor species were observed (Fig 3A). As expected, these double bands collapsed into a single band in the presence of a reducing agent (S5 Fig), except for the control that appeared partially resistant to reduction. For the positive cross-linked samples, approximately 50% of total receptor was cross-linked relative to CXCR4(D187C)-vMIP-II(W5C) control (Fig 3C, S4 Table), whereas for weaker crosslinks, this fraction was lower. In general, the fraction cross-linked was correlated with the amount of co-migrating chemokine (S4B Fig).

## Experiment-guided molecular modeling of the CXCR4 N-terminus interaction with CXCL12

Based on the resulting residue proximities, we generated atomic-resolution models of the CXCR4-CXCL12 complex (S1 Fig). For this, the fully flexible N-terminus of CXCR4 was subjected to biased probability Monte Carlo (BPMC) conformational sampling in internal coordinates [48], starting from random initial conformations and in the context of a flexible side-chain model of CXCL12. Three sulfated tyrosines (Y7, Y12, and Y21) were included in the receptor N-terminus to better understand their role in chemokine recognition [19,24,49]. In the course of sampling, weighted harmonic distance restraints were imposed between all receptor-chemokine residue pairs that demonstrated >65% cross-linking efficiency in flow cytometry, including the seemingly contradictory pairs, with weights calculated by a linear transformation of the cross-linking efficiencies (S6 Fig, S4 Table). Cross-linking efficiencies determined by pull-downs and Western blotting were not used in modeling to ensure that our cross-linking restraint set is large, unbiased, and methodologically uniform.

Following extensive sampling, the distal N-terminus extended into an antiparallel β-sheet with the CXCL12 β1-strand, mimicking the intermolecular packing of CXCL12 homodimers in the 3 lowest energy model conformations (S7 Fig). This geometry was consistently ranked first despite the cross-linking ambiguities across the different predicted interaction interfaces (Fig 2 and S6B Fig) and was also compatible with different CXCL12 backbone conformations (S8 Fig) clustered and selected based on our analysis of structural elements (S3 Table). To confirm mutual consistency between our approach and the resulting model, we introduced 2 explicit disulfides between residues pairs with high cross-linking efficiency by flow cytometry (K25C-E15C and G3C-L29C) and demonstrated that they can be accommodated with minor adjustments to the top-ranking geometry (S9 Fig).

## Detailed geometry of the complex between full-length CXCR4 and CXCL12

The best-scoring conformation of the complex featured the CXCL12 backbone geometry from PDB 3GV3 [50] (conformational cluster 2, S8 Fig and S3 Table). It was thus merged with the previously published partial (CRS2) CXCR4-CXCL12 model [43] to produce a full-length complex. The merged complex was subjected to local gradient minimization with restraints, which improved the interhelical packing in the TM domain as well as the packing of the

chemokine globular core against it. Finally, the proximal N-terminus of CXCL12 was refolded in the newly shaped TM binding pocket (CRS2) to complete the model (Fig 4A, S2 Data).

The resulting full-length model featured multiple newly predicted, energetically favorable polar residue interactions—for example, charge-charge interactions of CXCR4 CRS1 residues E26 with CXCL12 R47 (β3-strand), K25 with E15 (N-loop), D20 with K56 (C-terminal helix), and E15 with K43 (40s loop; Fig 4B, 4C and 4E). These interactions also persisted in at least 2 out of 3 top-ranking conformations of the partial complex (S7 Fig). Furthermore, the model explained the important role of sulfated tyrosines in chemokine recognition: extensive interactions were observed between the sulfated Y21 and CXCL12 N22 ($3_{10}$ helix), N44, and N45 (40s loop; Fig 4C and 4F). Although these proximities were somewhat supported by pull-down and Western blotting (Fig 3A and 3B), they were not evident from flow cytometry (Fig 2C) and, consequently, were not included as distance restraints in modeling (S4 Table). Thus, the identification of these interactions is not a trivial consequence of modeling restraints but rather an unbiased reflection of true binding preferences. These interactions are the key distinction between the lowest energy conformation and the other conformations (S7 Fig). Additionally, complementary charge-charge interaction clusters were predicted for the other sulfotyrosines in the receptor N-terminus (Y7 and Y12), the former with K24 and H25 in the chemokine β1-strand and the latter with K64 in its C-terminal helix (Fig 4B and 4F). All these interactions support the hypothesis that CXCL12 binding to CXCR4 is driven by electrostatics (S2 Fig) and explain the role of individual basic residues in the chemokine.

Our previous efforts to model CRS2 interactions of the CXCR4-CXCL12 complex resulted in positioning of the CXCL12 distal N-terminus in the minor subpocket, similar to the CXCR4-vMIP-II crystal structure that was used as a modeling template [6], with no occupancy of the major subpocket. However, the improved interhelical packing of the full-length model presented here allowed it to favorably and reproducibly accommodate the proximal N-terminus of CXCL12 in a new geometry, in which it occupies the major subpocket (Fig 4D and 4G and S10 Fig). This contrasts the geometry of complexes from CC [7] and CX$_3$C [8,9] families, in which the proximal N-terminus of chemokines occupies the minor subpocket instead (S10 Fig). The new geometry and the resulting occupancy of the major subpocket of CXCR4 by the proximal N-terminus of CXCL12 may be a defining feature for CXC receptor-chemokine selectivity against other subfamilies (CC and CX$_3$C; S10 Fig), unexplained by previous structural studies.

The final full-length model places in close vicinity those residue pairs that were demonstrated to cross-link, whereas non–cross-linking residues are mostly distant (S11 Fig). Whenever cross-link quantitation by flow cytometry was discordant with that by pull-down and Western blotting, the model tends to agree with the latter—for example, the pairs of CXCR4 S9/CXCL12 H17 (107.7% ± 16.3% efficiency by flow cytometry but only 29.8.1% ± 9.81% by pull-down and Western blotting), Y7/K27 (62.5% ± 5.3% versus 16.7% ± 2.0%), S9/L29 (47.9% ± 6.9% versus 13.2% ± 2.4%), and G19/K27 (76.0% ± 5.9% versus 18.4% ± 5.9%; Fig 3B and S4A Fig) are all distant in the lowest energy model, despite being present as restraints at the sampling stage. In fact, guided by these conflicting cross-links, the sampling procedure has identified potential intermediate steps in CXCR4-CXCL12 complex formation (S12 Fig). For example, because the CXCL12 H17C mutant cross-linked with most CXCR4 CRS0.5 and CRS1 residues, conformations in which the distal N-terminus resides in the vicinity of the N-loop were captured; these conformations may represent transient complex geometries that form before the CRS0.5 interaction is locked (Fig 4 and S12A Fig). An alternative explanation for these cross-links involves the CXCL12 dimer, in which the CXCR4 N-terminus in a fully extended conformation may reach CRS1 of the CXCL12 dimer partner (S12B Fig).

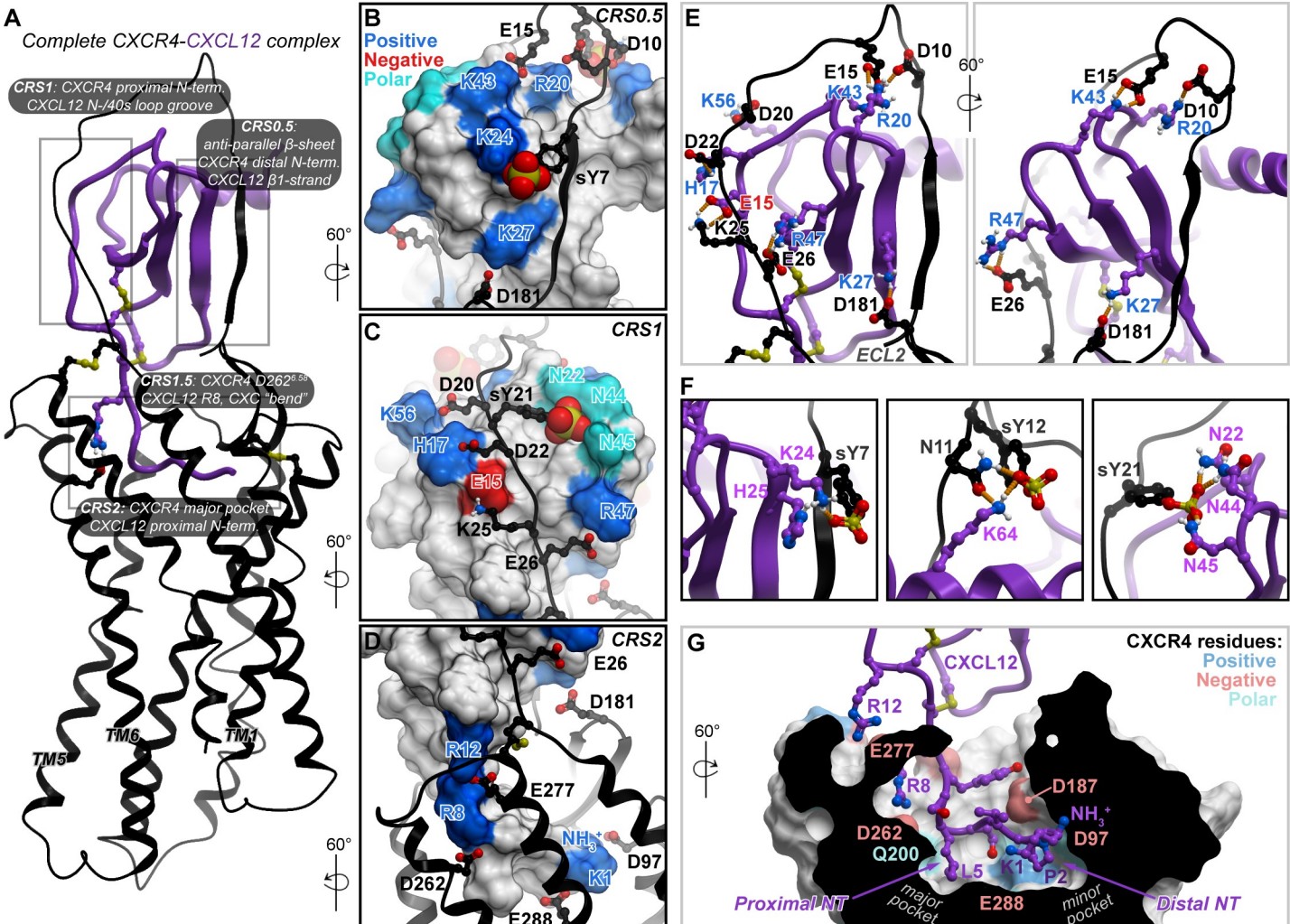

**Fig 4. Experiment-guided model of the full-length CXCR4-CXCL12 complex.** (A) An overview of the complete CXCR4-CXCL12 complex model. The receptor and CXCL12 are shown in black and purple ribbon, respectively. (B–E) The polar interactions across the various CRSs, which drive CXCL12 binding to the receptor. The receptor is shown in black ribbon and the chemokine in molecular surface mesh. Polar, negatively and positively charged side chains on the surface are colored in cyan, red, and blue, respectively, and the nonpolar surface is white. The chemokine residue labels are colored by their charge: negative (red) and positive (blue). In panel E, dominant polar interactions in CRS0.5 and CRS1 with hydrogen bonds are shown as orange dotted lines. (F) The predicted interactions between receptor sulfotyrosines and CXCL12 polar and charged residue clusters. (G) Predicted interactions of the CXCL12 N-terminus in the TM binding pocket of CXCR4. The binding of the proximal N-terminus of the chemokine in the major subpocket of the receptor is likely a defining feature for CXC receptor-chemokine complexes. The TM domain of the receptor is shown as a cut-away surface. Model coordinates are provided as S2 Data. CRS, chemokine recognition site; TM, transmembrane.

Finally, several weakly positive receptor-chemokine residue pairs could not be explained by either our best-scoring model geometry or by the alternative/transient geometries (S12 Fig). These pairs include CXCR4 M16C, Y21C, and S23C with [P2G]CXCL12 H25C and CXCR4 G19C with [P2G]CXCL12 K27C. We therefore sought to determine whether these proximities were consistent with other complex geometries proposed in the literature so far [6,25,51,52]. Previously published geometries were also incompatible with the offending cross-links, and, in fact, they are incompatible with the majority of cross-linking data generated here, including strong cross-links such as CXCR4 K25C with CXCL12 E15C, CXCR4 Y21C with CXCL12 H17C, and CXCR4 Y7C with CXCL12 H25C (S11 Fig). Therefore, in comparison with prior

models, the model generated here displayed marked improvement in recognizing the strongest experimental cross-links at shorter distances (S13 Fig).

### Validation of prospective predictions from the model in a functional assay

Our final model predicted several salt-bridge type interactions in CRS1 that were not identified by previously published models [25,51] and were also outside of the set of distance restraints used in the simulation. These included two salt bridges: one between CXCR4 E26 and CXCL12 R47 and another between CXCR4 D20 and CXCL12 K56 (Fig 5, S2 Data). Previous one-sided receptor mutagenesis studies have suggested a role for D20 and E26 in CXCL12 binding [53,54]; however, their functional role has not been verified. On the chemokine side, CXCL12 R47 mutations to Ala or Glu have been shown to reduce its potency towards CXCR4 activation [24]. However, the exact pairing between CXCR4 and CXCL12 residues in the complex has never been explored. Here, to validate model predictions, reciprocal charge reversal (also known as charge swap; S1 Table) mutagenesis was performed. Using a bioluminescence resonance energy transfer (BRET)-based β-arrestin-2 recruitment assay in human embryonic kidney 293T (HEK293T) cells, we found that wild-type (WT) CXCL12 displayed reduced potency and efficacy with the charge-reversal CXCR4 mutant E26R (pEC$_{50}$ 6.68 ± 0.07, Emax 87.9% ± 3.8%, $n$ = 3), relative to WT CXCR4 (pEC$_{50}$ 7.40 ± 0.04, Emax normalized to 100%; Fig 5A and 5B, Table 1). These defects were not a result of impairments in receptor surface expression (S14 Fig). On the chemokine side, the R47E mutation resulted in a partial agonist with strongly deteriorated potency and efficacy toward WT CXCR4 (pEC$_{50}$ 6.50 ± 0.10, Emax 88.23% ± 6.6%, $n$ = 3; Fig 5A and 5B, Table 1), consistent with previous studies [24]. However, when CXCR4(E26R) was stimulated by CXCL12(R47E), β-arrestin-2 recruitment efficacy was restored to WT levels, and potency was restored partially (pEC$_{50}$ 6.88 ± 0.08, Emax 101.8% ± 4.6%, $n$ = 3; Fig 5B), which confirms the predicted pairwise interaction of E26 and R47. This rescue of function by CXCL12(R47E) was specific to CXCR4(E26R) and was not observed with a proximal CXCR4 mutation, D20R, consistent with the lack of direct interaction between CXCR4 D20 and CXCL12 R47 in the model (Fig 5C and 5D).

Similar to the CXCR4 E26R, CXCR4 D20R demonstrated potency and efficacy deficits when stimulated with WT CXCL12 (pEC$_{50}$ 6.79 ± 0.08, Emax 78.54% ± 4.1%, $n$ = 4; Fig 5E and 5F, Table 1). However, on the chemokine side, the introduction of charge-reversal glutamic acid substitution on residue K56 did not lead to any detectable signaling deficits with WT CXCR4 (Fig 5E and 5F). Consequently, functional rescue of CXCR4 D20R with CXCL12 K56E was not attempted. With the highly positive electrostatic surface of CXCL12 (S2 Fig), it is conceivable that a single mutation does not result in a functional deficit. Overall, the pairwise charge-swap validation of the CXCR4 E26 and CXCL12 R47 interaction, as well as the loss of function with CXCR4 D20R mutation, provide strong prospective validation and support for our proposed CXCR4-CXCL12 geometry.

### Discussion

Many GPCRs have extended flexible N-termini, which are often posttranslationally modified and contribute to ligand binding and receptor signaling [55]. For chemokine receptors, the N-terminus is known to be a determinant of chemokine affinity and to contribute to specificity [14,17]; it is also increasingly being appreciated for its role in signaling efficacy [18,22,23]. However, because the distal N-termini of receptors are invariably unresolved in receptor-chemokine crystal structures, the structural basis for their regulation of chemokine affinity and signaling remains elusive.

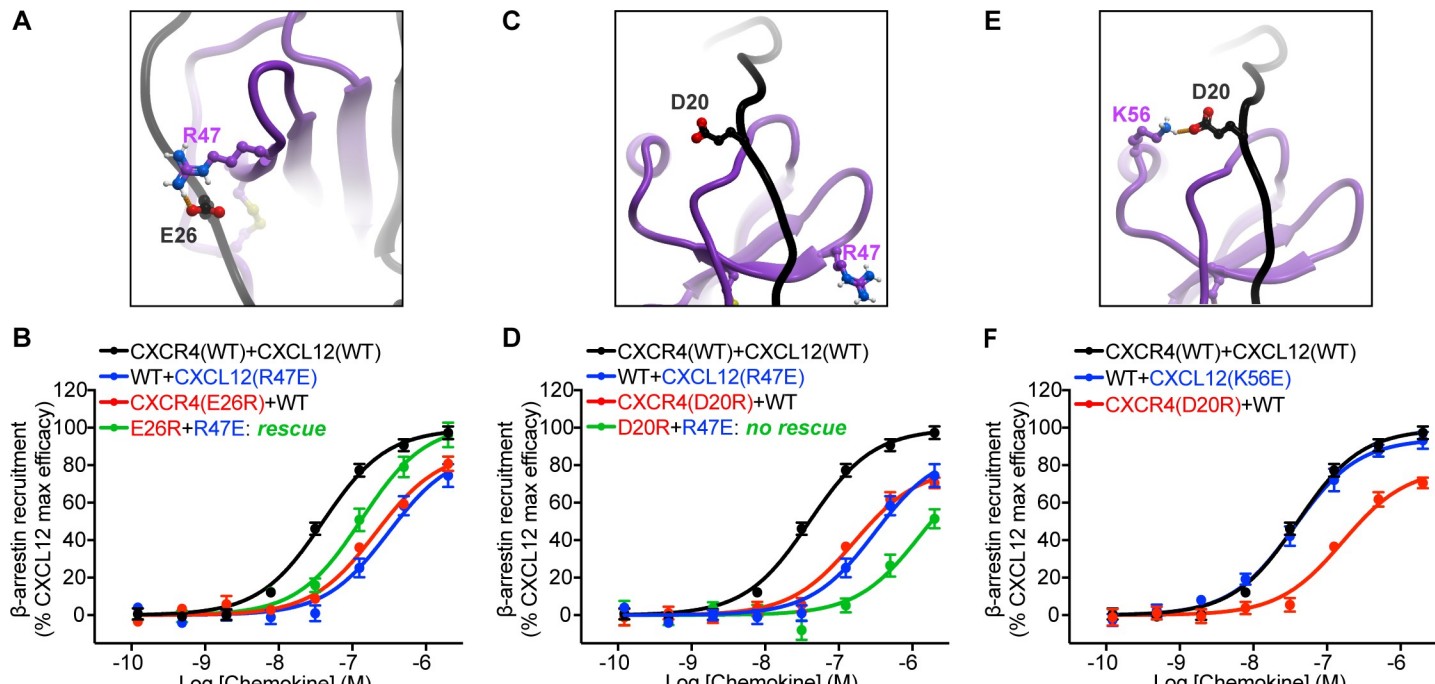

**Fig 5. Validation of predicted interactions in CRS1 via loss-of-function and reciprocal charge-reversal substitutions in a BRET2-based β-arrestin recruitment assay.** (A) The predicted interaction between CXCR4 E26 and CXCL12 R47. (B) Individual charge reversals (E26R in CXCR4 or R47E in CXCL12) lead to substantial deficits in both potency and efficacy of β-arrestin recruitment; however, when these mutants are combined, the deficits are partially rescued. (C) According to the model, CXCR4 D20 and CXCL12 R47 are located 15.8 Å away from each other and do not interact. (D) Combining CXCR4(D20R) and CXCL12(R47E) does not rescue signaling deficits caused by each of these mutations individually. (E) The predicted interaction between CXCR4 D20 and CXCL12 K56. (F) Unlike CXCR4(D20R), the K56E mutation in CXCL12 does not cause any detectable signaling defects, therefore, functional rescue was not attempted. Data represent at least $n$ = 3 independent biological replicates. The mean and SEM is reported for each point. WT curves were obtained in parallel with each respective mutant. The underlying numerical data for each figure panel can be found in S1 Data. BRET, bioluminescence resonance energy transfer; CRS, chemokine recognition site; WT, wild type.

Previous efforts to understand receptor-chemokine interactions have been predominantly "single-sided," that is, directed at characterization of binding surfaces of individual partners rather than pairwise interaction geometry (S1 Table). On the chemokine side, most of these studies established the N-loop/40s loop cleft as the key receptor binding interface [24,26,30,31], whereas some also pointed toward the involvement of CXC chemokine β1-strands. For example, in transferred cross-saturation NMR experiments, several CXCL12 β1-strand residues (L26, I28 and L29) displayed enhanced signal reduction ratios in the

**Table 1. Pharmacological parameters obtained in β-arrestin-2 recruitment assays.**

| pEC50 (Emax) | CXCL12(WT) | CXCL12(R47E) | CXCL12(K56E) |
|---|---|---|---|
| CXCR4(WT) | 7.40 ± 0.04 (normalized to 100%) | 6.50 ± 0.10[a] (88.23% ± 6.6%) | 7.44 ± 0.06 (94.1% ± 2.9%) |
| CXCR4(E26R) | 6.68 ± 0.07[a] (87.9% ± 3.8%[b]) | 6.88 ± 0.08[c] (101.8% ± 4.6%[d]) | ND |
| CXCR4(D20R) | 6.79 ± 0.08[a] (78.54% ± 4.1%[b]) | 5.89 ± 0.24[a] (85.22% ± 22.9%[be]) | ND |

[a]pEC50 significantly different from CXCR4(WT)-CXCL12(WT) according to the extra sum-of-squares F test ($p < 0.05$).

[b]Emax significantly different from CXCR4(WT)-CXCL12(WT) according to the extra sum-of-squares F test ($p < 0.05$).

[c]pEC50 significantly different from CXCR4(WT)-CXCL12(R47E) according to the extra sum-of-squares F test ($p < 0.05$).

[d]Emax significantly different from CXCR4(E26R)-CXCL12(WT) according to the extra sum-of-squares F test ($p < 0.05$).

[e]Curve incomplete and therefore Emax impossible to fit accurately.

**Abbreviations:** ND, not determined; pEC50, -$\log_{10}$ half maximal effective concentration; WT, wild type

presence of detergent-solubilized CXCR4 [56]. Similarly, in a radiolytic footprinting study, detergent-solubilized ACKR3 protected the same CXCL12 residues from radiolytic oxidation [34]. Perturbations of the CXCL8 β1-strand were also reported upon binding of soluble or membrane-bound CXCR1 N-terminus in solution NMR [26,57], and β1-strand mutations weakened the affinity and potency of CXCL8 towards CXCR1/2 [58,59].

To deduce the geometry and hence the structural basis for these phenomena, a series of complementary NMR studies were conducted; in these studies, isolated N-terminal peptides rather than full-length receptors were used. This approach bypasses challenges of purifying stable receptors in a membrane-like environment and can often capture dominant intermolecular proximities; however, the resulting interaction geometries often conflict with each other [24,25,30,49,60] and are also hard to reconcile with the receptor TM domains [32,33]. Recently, two independent studies postulated direct binding of receptor distal N-termini to the β1-strand of CXCL12: an NMR study [25] utilized pairwise proximities from intermolecular nuclear Overhauser effects did not involve the receptor TM domain, and yielded a parallel β-sheet geometry, whereas our study [34] involving a full-length homologous receptor ACKR3 but only one-sided interaction evidence by radiolytic footprinting proposed an antiparallel β-sheet.

The present work resolves these ambiguities. Using disulfide cross-linking we mapped, in a comprehensive and unbiased manner, pairwise residue proximities between the N-terminus of CXCR4 and [P2G]CXCL12, hence collecting unique interaction geometry information unattainable by the traditional "single-sided" approaches. Importantly, our effort was undertaken using a full-length receptor, hence avoiding artifacts caused by the absence of the TM domain. As a result we not only successfully recapitulated the CRS1 consistent with earlier NMR studies, models, and crystal structures of homologous receptors but also established the geometry of the novel CRS0.5, in which the distal N-terminus of CXCR4 extends into an antiparallel β-sheet with the β1-strand of CXCL12. The cross-linking–derived proximities were combined with molecular modeling to produce a complete atomic-resolution model of the complex between CXCL12 and full-length CXCR4, which was then validated prospectively both through loss-of-function and reciprocal charge-reversal mutagenesis (the latter an orthogonal experimental technique for testing pairwise residue interactions [S1 Table and the work by Stephens and colleagues [22]]).

The model, and in particular the novel CRS0.5 interface, reveal unique structural principles of CXC receptor-chemokine recognition, regulation, and signaling, as well as the basis of selectivity against other subfamilies. For example, unlike in the CC chemokine case [61], dimerization and receptor binding are not mutually exclusive for CXC chemokines [6]: the dimeric forms of such chemokines can activate receptors, albeit with affinity and pharmacology that are different from the monomers [24,62–64]. Because the receptor-chemokine interaction in CRS0.5 mimics and competes with CXC chemokine dimerization, it provides a structural basis for these differences. Moreover, CRS0.5 also explains the previously observed intriguing effects of β1-strand mutations on CC versus CXC chemokine selectivity, in which CC-mimicking mutations in the CXCL8 β1-strand endowed it with CC chemokine characteristics, including monocyte chemotactic activity and CCR1 binding [58,59], whereas reciprocal mutations bestowed neutrophil chemotactic activity (intrinsic to CXC chemokines) onto CCL2 [65]. Finally, the selectivity insights from the model extend beyond CRS0.5: we also propose the basis for unique engagement of the proximal chemokine N-terminus in the major subpocket of the receptor, in which the conserved arginine residue preceding the chemokine CXC motif (R8 in CXCL12) is coordinated by the conserved acid in the receptor TM6 position 6.58 (D262$^{6.58}$ in CXCR4, Fig 4G). This engagement depends on a unique CXC-motif-dependent "bend" [6] in the chemokine N-terminus [6,60] and may be a distinguishing feature of CXC

complexes (S10 Fig). In the case of CXCR4-CXCL12 it is consistent with reports of deleterious effects of D262[6.58] and R8 mutations, respectively [22,25,33]. It is also consistent with earlier observations of the critical role of the "ELR" (Glu-Leu-Arg) motif in other, neutrophil-activating CXC chemokines [66,67], as well as the corresponding TM6 residue of neutrophil chemotactic CXC receptors [68,69] (D265[6.58] in CXCR1). Both the chemokine N-terminal "bend" and the conserved CXC-specific recognition determinants are absent in the CC and CX$_3$C complexes [7–9] (S10 Fig) and thus contribute to intersubfamily selectivity.

In fact, the CXCR4-CXCL12 complex may be unique compared to other CXC subfamily receptor-chemokine pairs. First, it extensively relies on the chemokine distal N-terminus interactions with the minor subpocket [10,33]. This is in contrast to other, neutrophil-specific CXC complexes that tolerate N-terminal truncations up to the "ELR" motif in the chemokine with only minor impairment of signaling [66,70]. Additionally, CXCR4 features a basic amino acid (R30) 2 residues downstream of the N-terminal cysteine, which is a characteristic of CC rather than CXC receptors [60]. We hypothesize that these features allow CXCR4 to bind, with high affinity, to the virally encoded CC chemokine vMIP-II [6], which itself mimics CXC chemokines by having a unique arginine (R7) in the proximal N-terminus, highly important for receptor binding [71] (S10 Fig). Therefore, the features identified by the present CXCR4-CXCL12 model explain not only CXC versus CC/CX$_3$C selectivity but also the rare and seemingly paradoxical coupling between a CXC receptor and a CC chemokine.

Our results suggest the structural basis by which tyrosine sulfation of the CXCR4 N-terminus improves its binding affinity for CXCL12. Corroborating our predictions, earlier NMR studies of CXCL12 with sulfotyrosinated receptor N-terminal peptides identified sY21 to be positioned similarly in the CXCL12 N-loop/40s loop groove and interacting with N22 and N46 (compared to the cluster of N22, N44 and N45 in our model, Fig 4C) [24,49]. The predicted interactions of sY21 with an asparagine cluster of CXCL12 are reminiscent of CCR5 sulfotyrosine recognition by a similar cluster of the HIV envelope glycoprotein gp120 [72], suggesting that asparagine clusters may constitute preferred sulfotyrosine binding motifs. By contrast with sY21, the placement of other sulfotyrosines in earlier NMR-derived CXCR4-CXCL12 models is incompatible with the spatial constraints imposed by the receptor TM domain and the structurally determined receptor-chemokine architecture [6,33]. This reiterates the importance of studying receptor-chemokine interactions in the context of the full-length receptor.

Interestingly, ambiguities in the cross-linking data allowed for several alternative CXCR4-CXCL12 complex geometries, among which we identified the most probable and favorable one by flexible molecular docking in the presence of weighted local distance restraints. The locality ensured that the maximum number of restraints was satisfied in the final model, hence overcoming the ambiguous signal across the proposed interaction sites. This approach is conceptually similar to nuclear Overhauser effect-based distance restraints in NMR structure calculations [30]. Importantly, alternative geometries captured in the sampling procedure may represent biologically relevant intermediates (S12 Fig). A limitation in our modeling approach is that the orientation of the chemokine globular core with respect to the receptor TM domain, and activation state of the latter, originated from the CXCR4-vMIP-II antagonist crystal structure and were only locally minimized in the course of full model assembly, whereas more recent structures demonstrate substantial variability in both [7,8]. In future studies, molecular dynamics will be required to thoroughly sample the system and identify more precisely the positioning of the CXCL12 globular core with the bound receptor N-terminus relative to the rest of the receptor.

To detect and quantify covalent receptor-chemokine complexes, we developed a medium-throughput flow cytometry–based screening approach. The assay is performed in a multiwell

format and does not require sample lysis, extraction, or purification, nor does it depend on unnatural amino acids or photoaffinity labeling, hence contrasting and improving on previous cross-linking studies of GPCR-protein complexes [33–35,73,74]. Nevertheless, there are caveats inherent to disulfide cross-linking, resulting in several cross-links that could not be explained by any of the proposed CXCR4-CXCL12 complex geometries. First, cysteine mutations may lead to protein destabilization, misfolding (for example, CXCL12 L26C and I28C mutations), or loss of binding. Additionally, introducing additional cysteines in the already cysteine-rich environment of chemokines and receptors increases the possibility of intramolecular or intermolecular disulfide shuffling upon folding or complex formation [35].

Despite these limitations, the comprehensive, systematic, and unbiased manner in which disulfide cross-linking was performed and interpreted here allowed us to distinguish native residue proximities with high confidence and accuracy, confirmed by prospective model-guided mutagenesis and charge-swap experiments. Altogether, this effort resulted in the determination of the geometry of the new CRS0.5 interface, construction of the most unifying and complete high-resolution model of the prototypical CXC receptor-chemokine complex, and a structural understanding of key principles of receptor-chemokine recognition and selectivity that were not apparent from prior structural studies of other chemokine subfamily complexes. Insights from the model will guide the rational development of chemokine-targeting small molecules and biologics with new pharmacology and selectivity profiles.

## Materials and methods

### Molecular cloning

For cross-linking studies in *Sf9* cells, cysteine mutations were introduced into the N-terminally Flag-tagged, C-terminally His-tagged CXCR4-T4 lysozyme fusion construct in pFastBac1 (CXCR4-T4L) [32,33] using QuikChange mutagenesis (Agilent, Santa Clara, CA). Cysteine mutations were also generated for the antagonist CXCL12 variant [P2G]CXCL12, with its native signal sequence and a C-terminal HA tag (YPYDVPDYA), subcloned into pFastBac1 ([P2G]CXCL12-HA). The CXCR4(D187C)-T4L and vMIP-II(W5C) mutants and the C-terminally tagged ACKR3-Flag construct were previously described [6,34]. For recombinant chemokine production, a pET21a-based construct of human CXCL12α (residues 22–89) lacking the endogenous signal peptide and preceded by an enterokinase recognition site and His8 tag [34,75] was used as the WT construct; CXCL12 R47E and K56E mutations were introduced into this construct using QuikChange mutagenesis (Agilent, Santa Clara, CA). For pharmacological assays in HEK293T cells, the WT CXCR4 fused with Renilla luciferase 3 (Rluc3) on the C-terminus and β-arrestin-2 fused N-terminally to GFP10 constructs in pcDNA3.1+ were generously provided by Nikolaus Heveker (Université de Montréal, Montréal, QC, Canada). An HA tag following the CXCR4 start codon, as well as D20R and E26R mutations were introduced by QuikChange mutagenesis (Agilent, Santa Clara, CA).

### Expression and production of recombinant chemokines in *E. coli*

Chemokine expression and purification protocol was adapted from [34]. Chemokine-encoding plasmids were transformed into BL21(DE3)pLys competent cells. Cells were grown at 37˚C with shaking at 180 rpm to an optical density of 0.6 to 0.7 in Luria-Bertani medium, and protein expression was induced by addition of 0.5 mM isopropyl β-D-1-thiogalactopyranoside. Following 6 h of induction at 37˚C, cells were harvested by centrifugation. Cell pellets were resuspended in lysis buffer (50 mM Tris, 150 mM NaCl [pH 7.5]) with 1 μg/mL DNaseI and lysed by sonication. The samples were centrifuged (12,000*g* for 15 min), and the insoluble pellets containing chemokine inclusion bodies were collected. The pellets were dissolved in

Equilibration buffer (50 mM Tris, 6 M guanidine-HCl, 50 mM NaCl [pH 8.0]), sonicated, and the supernatant (containing chemokine) was collected by centrifugation (12,000$g$ for 15 min). The supernatant was loaded onto Ni-NTA agarose resin (Qiagen, Hilden, Germany), washed with Wash buffer (50 mM MES, 6 M guanidine-HCl, 50 mM NaCl [pH 6.0]), and the chemokine was eluted with Elution buffer (50 mM acetate, 6 M guanidine-HCl, 50 mM NaCl [pH 4.0]). The eluate was brought to pH > 7.0 with NaOH, reduced with 4 mM DTT for 2 h at room temperature, diluted 10-fold into Refolding buffer (50 mM Tris, 500 mM arginine-HCl, 1 mM EDTA, 1 mM glutathione disulfide [pH 7.5]), and incubated for 60 min at room temperature with moderate stirring. Following dialysis of the refolding mixture (20 mM Tris, 50 mM NaCl [pH 8.0]), the dialysis product was cleared of precipitant by centrifugation (8,000$g$ for 15 min) and concentrated to approximately 10 to 20 mL with a 3 kDa cutoff Amicon centrifugal filter unit (MilliporeSigma, Burlington, MA). After concentration, 2 mM CaCl$_2$ was added and the His$_8$ tag was cleaved through the addition of 8 to 16 U/mL enterokinase (New England Biolabs, Ipswich, MA) for 2 days at 37˚C. After cleavage, the mixture was loaded onto Ni-NTA resin, and tag-free CXCL12 was eluted with Wash buffer. The eluate was bound to a reversed-phase C18 HPLC column (Vydac; buffer A: 0.1% trifluoracetic acid; buffer B: 0.1% trifluoroacetic acid; 90% acetonitrile) and eluted by linearly increasing the buffer B concentration from 33% to 45%. The resulting recombinant chemokine was lyophilized and stored at −80 ˚C until use.

## Production of receptor and chemokine viruses for co-expression in *Sf9* cells

*Sf9* insect cells were maintained in ESF 921 media (Expression Systems, Davis, CA) in vented Erlenmeyer flasks (Corning Inc, Corning, NY) at 27˚C with shaking at 130 rpm.

Receptor and chemokine baculovirus stocks were produced using the Bac-to-Bac Baculovirus Expression System (Invitrogen, Carlsbad, CA). Briefly, 2.5 mL of *Sf9* cells at a density of $1.2 \times 10^6$ cells/mL were transfected with 5 μL of recombinant bacmid containing the gene of interest using 3 μL of X-tremeGENE HP DNA transfection reagent (Roche, Basel, Switzerland) and 100 μL of transfection medium (Expression Systems, Davis, CA) and incubated for 96 h with 300 rpm shaking at 27˚C. Following cell centrifugation at 2,000$g$ for 10 min, the P0 virus (supernatant) was isolated, and 400 μL of P0 virus was immediately added to 40 mL of *Sf9* cells at a density of 2 to $2.6 \times 10^6$ cells/mL for generation of P1 virus. Cells were incubated for 48 h with 130 rpm shaking at 27˚C and centrifuged at 2,000$g$ for 10 min; the supernatant containing the P1 virus was harvested and stored at 4˚C until further use.

Viral titers were calculated in infectious units (IU)/mL as previously described [76]. Briefly, 20 μL of serial P1 virus dilutions (1 in 250, 500, 1,000, and 2,000) was added to 100 μL of *Sf9* cells at a density of $1.2 \times 10^6$ cells/mL in a 96-deep well block and incubated for 18 h with 300 rpm at 27˚C. The percentage of cells infected by virus was quantified by flow cytometry following staining of 10 μL of cells with 10 μL of phycoerythrin-conjugated anti-gp64 antibody (Expression Systems, Davis, CA, catalog #97–201) for 20 min in the dark at 4˚C. The anti-gp64-phycoerythrin antibody was prediluted 1:100 in Tris-buffered saline (TBS; 50 mM Tris [pH 7.5], 150 mM NaCl) and 4% BSA. Following incubation, samples were made up to 200 μL with TBS+4% BSA and analyzed using a Guava benchtop mini-flow cytometer (EMD Millipore). Virus titer was quantified as an average of (total cell number × %gp64 staining × virus dilution factor)/20 μL (volume of inoculum).

## Detection of covalent receptor-chemokine complexes by flow cytometry

Flow cytometry used to initially detect disulfide cross-linked complexes on the cell surface was adapted from Kufareva and colleagues [35]. Briefly, 2.5 mL of *Sf9* cells at a density of 2 to

$2.6 \times 10^6$ cells/mL in a 24-deep well block were co-infected with receptor and chemokine P1 virus, each at a multiplicity of infection (IU/cell) of 8. Cells were incubated with 300 rpm shaking at 27˚C. After 48 h, 10 μL of cell samples were transferred into a 96-well assay plate and incubated for 20 min in the dark at 4˚C with 2 μL 7-amino-actinomycin D viability staining solution (eBioscience, Santa Clara, CA, catalog #00-6993-50) and 10 μL anti-HA FITC-conjugated antibody (Sigma-Aldrich, catalog #H7411), the latter prediluted 1:100 in TBS+4% BSA. Following incubation, samples were made up to 200 μL with TBS+4% BSA and analyzed using a Guava benchtop mini-flow cytometer (EMD Millipore). In all experiments, the previously characterized CXCR4(D187C)-vMIP-II(W5C) cross-linked pair [6] was used as a positive control. For data analysis, the geometric mean fluorescence intensity (GMFI) value of the live cell population was used. Cross-linking efficiency by flow cytometry was determined as GMFI for the receptor-chemokine pair in relation to CXCR4(D187C)-vMIP-II(W5C), which was set to 100%. All flow cytometry data analysis was performed using FlowJo software version 10.3 (FlowJo LLC, Ashland, OR).

## Metal affinity pull-down of covalent receptor-chemokine complexes from *Sf*9 cells

To confirm the receptor-chemokine cross-links detected by flow cytometry, metal affinity pull-downs were performed using the His-tag on the receptor, with subsequent receptor and chemokine detection by SDS-PAGE and quantification by Western blotting. A total of 40 mL of *Sf9* cells at a density of 2 to $2.6 \times 10^6$ cells/mL were infected with receptor and chemokine virus, each at an multiplicity of infection of 5, and incubated for 48 h with 130 rpm shaking at 27˚C. Biomass was harvested by centrifugation (2,000*g* for 15 min) and stored at −80˚C until further use.

Cell pellets were thawed and resuspended in hypotonic buffer (10 mM HEPES [pH 7.5], 20 mM KCl, 10 mM MgCl₂, 1 × cOmplete EDTA-free protease inhibitor [Roche]). Purified membranes were prepared by 3 rounds of dounce homogenization (approximately 40 strokes per round) and centrifugation of the cell suspension at 50,000*g* for 30 min at 4˚C. Following the first round of homogenization, membrane pellets were resuspended in a high salt buffer (10 mM HEPES [pH 7.5], 10 mM MgCl₂, 20 mM KCl, 1 M NaCl, and 1 × cOmplete EDTA-free protease inhibitor) which was also used during the subsequent rounds. After the last round of centrifugation, membranes were resuspended and homogenized in hypotonic buffer supplemented with 30% glycerol (v/v) and flash-frozen for storage at −80˚C until further use.

The purified membranes were thawed and iodoacetamide (2 mg/mL) added. Membranes were mixed with an equal volume of 2 × solubilization buffer (100 mM HEPES [pH 7.5], 800 mM NaCl, 1.5% n-dodecyl-ß-D-maltopyranoside [DDM, Anatrace], 0.3% cholesteryl hemisuccinate [CHS, Sigma, St. Louis, MO]) for 4 h at 4˚C and centrifuged at 25,000*g* for 30 min. The supernatant was incubated with TALON IMAC resin (Clontech, Mountain View, CA) and 25 mM of imidazole overnight. The samples were centrifuged at 350*g* for 5 min, and the resin was transferred to gravity flow columns. The resin was washed with 20 column volumes of Wash Buffer 1 (25 mM HEPES [pH 7.5], 400 mM NaCl, 10% glycerol, 0.1/0.02% DDM/CHS, 10 mM imidazole) followed by 10 column volumes of Wash Buffer 2 (25 mM HEPES [pH 7.5], 400 mM NaCl, 10% glycerol, 0.025/0.005% DDM/CHS, 10 mM imidazole). Complexes were eluted with 3 column volumes of elution buffer (Wash Buffer 2 with 250 mM imidazole), and the eluted protein was exchanged into Wash Buffer 2 without imidazole using 0.5 mL 100 kDa molecular weight cutoff spin concentrators.

## Characterization of covalent receptor-chemokine complex formation by SDS-PAGE and Western blotting

Protein samples from metal affinity pull-down experiments were analyzed for formation of cross-linked complexes by SDS-PAGE followed by Coomassie staining or Western blotting [35]. Samples (1 μg total protein) were loaded onto nonreducing 12% Mini-PROTEAN TGX precast gels (Bio-Rad, Hercules, CA), and molecular weight shifts and relative band intensity used to detect the presence and relative abundance of the cross-linked complex. For Western blotting, gels were transferred onto nitrocellulose membranes (Bio-Rad, Hercules, CA, catalog #1704270) using the Trans-Blot Turbo transfer system, and blocked with 5% skim milk over-night at 4°C. The Flag-tagged receptor was detected using mouse anti-Flag M2 primary antibody (1:10,000 dilution; Sigma Aldrich, St. Louis, MO, catalog #F3165) and IRDye 680RD-conjugated donkey anti-mouse IgG secondary antibody (1:20,000 dilution; LI-COR Biosciences, Lincoln, NE, catalog #926–68072). HA-tagged [P2G]CXCL12 was detected using a rabbit anti-HA primary antibody (1:10,000 dilution; Sigma Aldrich, St. Louis, MO, catalog #SAB4300603) and IRDye® 800CW-conjugated goat anti-rabbit IgG (1:20,000 dilution; LI-COR Biosciences, Lincoln, NE, catalog #926–32211) secondary antibody. Primary antibodies were incubated simultaneously for 1 h at room temperature, followed by the secondary antibodies for a further 1 h at room temperature. Primary and secondary antibodies were diluted in TBS with 0.5% Tween-20 and 5% skim milk or 1% BSA, respectively. Membranes were imaged using the Odyssey IR imaging system (LI-COR Bioscience, Lincoln, NE). For reducing conditions, samples were incubated with 100 mM DTT for 30 min at room temperature prior to loading onto gels.

To quantify the amount of cross-linked chemokine, a rectangle was drawn on the anti-HA Western blot that captures the cross-linked chemokine band, and IR800 signal intensity within the rectangle was recorded as reported by ImageStudioLite (LI-COR Biosciences, Lincoln, NE). This value was normalized to CXCR4(D187C)-vMIP-II(W5C), where its IR800 signal was set to 100%, and used as the cross-linking efficiency by Western blot. To calculate the percentage of cross-linked receptor, the same chemokine rectangle was used to capture the upper chemokine–cross-linked receptor band, and a second rectangle was drawn to capture total receptor. The percent cross-linked receptor was determined by upper band IR680 signal/total receptor IR680, where IR680 reports the receptor Flag tag. The area of the rectangles was kept consistent for all samples within a single Western blot.

## Molecular modeling

All molecular modeling was carried out using ICM [48] version 3.8–6 or higher (Molsoft LLC, San Diego, CA), and the previously published incomplete model (residues P27-F304) in complex with CXCL12 [6]. The N-terminal peptide M1-P27 was constructed ab initio, tyrosines Y7, Y12, and Y21 were sulfated [19,24], and residue P27 was tethered to its position in the earlier model. Weighted local harmonic distance restraints were imposed between Cβ atoms (or Cα atoms for Gly) of those CXCR4 and CXCL12 residues that displayed >65% cross-linking efficiency by flow cytometry. The conversion of cross-linking efficiencies into weights involved a linear normalizing transformation so that the weights ranged from 0.5 to 5 (S4 Table). The mechanism of local distance restraints favors proximal restrained atom pairs, with restraint energy gradually becoming weaker (less favorable) and approaching an asymptote as the distance between the 2 atoms grows, rather than increasing indefinitely (S15 Fig). The parameters of the energy function were chosen to favor distances below approximately 4 Å (S15 Fig), because this is representative of Cβ-Cβ distances in disulfide bonds. The explicit sampling system included a fully flexible CXCR4 N-terminus (residues 1:27, sulfated at Y7, Y12, and Y21)

and the full-atom model of CXCL12 with fixed backbone and flexible side chains. The objective (energy) function included full-atom van der Waals term calculated using the Lennard-Jones potential and capped at 7 kcal/mol (the so-called soft van der Waals energy), hydrogen bonding term, electrostatics, torsional strain, tethers, and a medium-strength penalty (cnWeight = 5) for distance restraints (ICM energy/penalty term "cn", short for "contacts"). To account for backbone flexibility of CXCL12, parallel simulations were run with alternative backbone conformations; for this, crystal structures of CXCL12 were clustered by the geometry of the 30s and 40s loop, and the highest resolution structure was selected as the representative conformation for each cluster (Fig 1C, S3 Table). Each system was sampled by BPMC [48] for $3 \times 10^7$ steps, and the lowest energy conformations were stored.

The top-ranking conformation of the CXCR4 N-terminus with CXCL12 was merged with the rest of the receptor [6] to build a complete model of the complex. The globular core of the chemokine was superimposed onto that of the partial model, after which contiguous CXCR4 and CXCL12 polypeptide chains were generated ab initio and threaded through the respective parts of the overlay, recapitulating the CRS0.5 and CRS1 interactions determined above, as well as previously published CRS2 interactions [6]. The model was then subjected to global refinement with restraints. For this, 264 harmonic global distance restraints were imposed in place of the inter- and intramolecular hydrogen bonds: the intramolecular bonds were extracted from the separate crystal structures of the 2 proteins (CXCR4: PDB 3ODU [32] and 4RWS [6]; CXCL12: PDB 3GV3 [50]), whereas the intermolecular hydrogen bonds were inferred from the CRS interfaces from the model in the prior step. Three additional hydrogen bond–compatible global distance restraints were added: one between CXCR4 E277 and CXCL12 R12 that were previously predicted and recently experimentally proved to directly interact [22], another between CXCR4 Q200 and D262, to promote closure of the gap between TM helices 5 and 6 that is observed in CXCR4 crystal structures, and the third between CXCR4 D181 and CXCL12 K27, which were proximal in the original model but outside of the hydrogen bonding distance. The last restraint promoted slight re-orientation and better packing of the globular core of CXCL12, together with the bound CXCL12 N-terminus, with respect to CXCR4 TM domain. Next, the system was subjected to a refinement protocol consisting of global BPMC sampling of receptor and chemokine residue side chains, with each sampling step followed by 500 steps of local gradient minimization of the entire system (backbone and side chains) in internal coordinates. The objective (energy) function included full-atom soft van der Waals term capped at 20 kcal/mol, hydrogen bonding term, electrostatics, torsional strain, surface energy, and the penalty for the distance restraints. After $10^6$ minimization steps, 2 loops, one involving CXCL12 residues 3–7, and another CXCR4 residues 224–236 ($3^{rd}$ intracellular loop), were refolded using the ICM loop modeling toolkit. Briefly, this involved finding, by BPMC, the optimal conformation for each loop in the context of the rest of the model, with the loop represented as an isolated peptide with termini tethered to their positions in the model, and the rest of the model represented as energy potentials (van der Waals, electrostatics, hydrogen bonding, and surface energy) precalculated on a 0.5 Å 3D grid [77]. The best conformations were merged into the full-model of the complex, and the model subjected to additional $2 \times 10^6$ steps of the above refinement protocol, to ensure that the energy close to the local minimum is achieved.

## BRET2 β-arrestin recruitment assay

β-arrestin-2 recruitment was measured with BRET2 assay [78] as described previously [34]. Briefly, HEK293T cells were seeded in tissue-culture-treated 6-well plates at $7 \times 10^5$ cells cells per well in Dulbecco's Modified Eagle Media + 10% fetal bovine serum. Next day, cells were

co-transfected with 50 ng/well WT or mutant HA-CXCR4-Rluc3 and 2,400 ng/well GFP10-β-arrestin-2 using TransIT-X2 transfection reagent (MirusBio, Madison, WI), according to the manufacturer's recommended protocol. Two days after transfection, cells were washed with PBS, resuspended in PBS + 0.1% D-glucose (BRET buffer), and diluted to 1 to $1.5 \times 10^6$ cells/mL. Cells were aliquoted at $1.5 \times 10^5$ cells per well into a white, clear bottom, tissue culture treated 96-well plate (Corning Inc, Corning, NY), and the plate was incubated for 30 min at 37°C, after which GFP10-β arrestin-2 fluorescence levels were measured with the TriStar LB 941 plate reader (Berthold Biotechnologies, Baden-Württemberg, Germany). Next, recombinant WT or mutant CXCL12 was added to each well at indicated final concentrations and incubated for 10 min, after which coelenterazine-400A (also known as DeepBlueC) was added to a final concentration of 5 uM immediately before measuring the emission at both 410 and 515 nm using the TriStar LB 941 plate reader. BRET ratios (emission at 515 nm/emission at 410 nm) were calculated, and three-parameter agonist concentration response curve fitting was carried out using GraphPad Prism (GraphPad Software Inc, San Diego, CA). BRET2 values were baseline-corrected and normalized to CXCR4(WT)-CXCL12(WT) levels within each experiment. The extra sum-of-squares F test was used to compare curve fit parameters and determine statistically significant differences between WT and mutant receptors and chemokines. Results are listed in text as $pEC_{50}$/Emax ± the standard error for that parameter, when normalized values from at least 3 independent experiments were fit together.

### Detection of CXCR4 surface expression by flow cytometry

The cell surface expression of WT HA-CXCR4-Rluc3 and mutants was monitored by flow cytometry using the same cells as in the BRET experiments above. Cells resuspended in BRET buffer were aliquoted at $1.5 \times 10^5$ cells per well into a conical-bottom 96-well plate and spun down (250$g$ for 5 min). Supernatant was removed, and cells were resuspended in 50 μL of allophycocyanin-conjugated anti-CXCR4 antibody (clone 12G5, BD Biosciences, San Diego, CA, catalog number 560936) diluted 1:50 in FACS buffer (PBS + 0.5% BSA). Cells were stained in the dark for 45 min at 4°C, as recommended by the manufacturer. Following incubation, cells were washed 3 times with FACS buffer and fixed with a final concentration of 0.8% paraformaldehyde. Samples were analyzed using a Guava benchtop mini-flow cytometer (EMD Millipore), and cell surface expression was given by GMFI in the RED-R channel. The surface expression of CXCR4 mutants was normalized to WT CXCR4.

### Supporting information

**S1 Raw Images for Gels and Blots. Raw uncropped images of SDS-PAGE gels and Western blot membranes of Fig 3 and S5 Fig.**
(PDF)

**S1 Fig. Overview of the disulfide cross-linking–based strategy for determining the geometry of receptor-chemokine complexes in this study.**
(TIF)

**S2 Fig. The electrostatic surface of CXCL12.** The CXCL12 surface is strongly positively charged (blue). The electrostatic surface was calculated by the Rapid Exact-Boundary ELectrostatics (REBEL) method in ICM [79]. The black strokes highlight the predicted peptide interaction grooves on the CXCL12 surface: one between the N-loop and 40s loop, and another along the β1-strand backbone.
(TIF)

**S3 Fig. Validation of folding of [P2G]CXCL12 Cys mutants by detecting their noncovalent binding to ACKR3.** ACKR3 and [P2G]CXCL12 cysteine mutants were co-expressed in *Sf9* cells. Due to the slow off-rate of [P2G]CXCL12 with ACKR3, complexes readily detected on the cell surface are a proxy for mutant chemokine folding. All mutants except L26C, I28C, and L29C retain their ability to bind ACKR3. *n* = 4 independent biological replicates. The mean and SEM are reported for each point. The underlying numerical data for the figure can be found in S1 Data. *Sf9*, *Spodoptera frugiperda*.
(TIF)

**S4 Fig.  Positive relationship between the detection of the chemokine HA tag on the cell surface and in the pulled down cross-linked samples (A) and between the HA intensity and the fraction cross-linked receptor in the pulled down samples (B).** Flow cytometry was used for detection of the chemokine HA tag on the cell surface, whereas Western blot was used for pulled-down protein samples. Data represent mean and SEM of 3 or more independent replicates. The underlying numerical data for each figure panel can be found in S1 Data.
(TIF)

**S5 Fig. The covalent CXCR4-[P2G]CXCL12 complexes dissociate to separate single receptor and chemokine species under reducing conditions.** Western blot of pulled down combinations of CXCR4 and [P2G]CXCL12 cysteine mutants in the presence of 100 mM DTT. The Flag-CXCR4-T4L receptor and [P2G]CXCL12-HA chemokine were detected by LI-COR IRDye conjugated secondary antibodies on a single blot (emission wavelength of 680 nm and 800 nm, visualized in red and green, respectively). Emitted fluorescence detected at 800 nm and 680 nm from different bands of the Western blot is indicative of a dissociated receptor-chemokine complex. The figure is representative of *n* = 3 independent replicates.
(TIF)

**S6 Fig. The weighted distance restraints imposed between residue Cβ atoms (Cα for glycine) during molecular docking.** (A) Graphical representation of the experimentally derived local distance restraints imposed during the molecular docking simulations. Cross-interface restraints are not shown. Distance restraints are colored by a gradient of blue according to their experimentally determined strength. (B) The distance restraints are mapped onto 3 randomly selected starting conformations and the top ranked conformation of the receptor N terminus. Distance restraints are shown in dotted lines, colored by a gradient of blue as in panel A. The receptor N terminus and CXCL12 are shown in black and purple ribbon, respectively. The underlying values of the distance restraint weights are found in S4 Table.
(TIF)

**S7 Fig. The top 3 ranking conformations of the CXCR4 receptor N terminus.** The lowest energy conformations are distinct from the other conformations. The conformational stack was sorted by the energy of the system. Polar and charge interactions are shown in orange dotted lines. The receptor and CXCL12 are shown in black and purple ribbon, respectively. The TM domain is hidden for clarity. The underlying numerical data for the figure can be found in S1 Data. TM, transmembrane.
(TIF)

**S8 Fig. The proposed geometry of the receptor N terminus and the CRS0.5 interface is compatible with various CXCL12 backbone conformations.** The top-ranking conformation from each respective simulation is shown: in all cases, the receptor N terminus forms an interface with the CXCL12 β1-strand. The CXCL12 conformation PDB 3GV3 from Cluster 2 was selected for full-length complex assembly. The receptor is shown in black, and CXCL12 is

colored distinctly in each model. CRS, chemokine recognition site.
(TIF)

**S9 Fig. Structural context of the cross-linking approach.** Experimental cross-linking observed between the 2 pairs of CXCR4-CXCL12 residues (E15-K25 and L29-G3) can easily be accommodated structurally in our top-ranking model with minor changes to the conformation of the receptor N terminus.
(TIF)

**S10 Fig. Occupancy of the receptor major subpocket by the chemokine proximal N terminus defines chemokine receptor subfamily selectivity.** (A–C) Compared to Fig 4G, the proximal N terminus of CC and $CX_3C$ chemokines occupy the receptor minor subpocket. vMIP-II is unique among CC chemokines, containing an arginine, conserved in CXC chemokines, allowing it to partially occupy the top of the receptor major subpocket. (D) Overlay of the CC and $CX_3C$ chemokines determined in crystal structures, along with CXCL12. CXC chemokines have a pronounced N-terminal bend.
(TIF)

**S11 Fig. Previous published models of CXCR4-CXCL12 complex are incompatible with the cross-linking data.** Receptor-chemokine residue Cβ-Cβ (or Cα for Gly) distances were calculated for 3 models of the CXCR4-CXCL12 complex and projected onto a heat map for comparison with experimental crosslinking. (A) The model generated here; (B) the model published by Tamamis and Floudas [51]; (C) the model published by Ziarek and colleagues [25]. We note that the Tamamis and Floudas model was built prior to publication of the CXCR4-vMIP-II crystal structure, and that the Ziarek and colleagues model was informed by NMR of CXCL12 with an isolated N-terminal peptide of CXCR4. In the Ziarek and colleagues model, residue G3 was not modeled (dark gray in the heat map). Cβ-Cβ distances between residue pairs (CXCR4 K25-CXCL12 E15, Y21-H17, and Y7-H25) are shown in blue dotted lines, and their distances are given in Ångstroms. The receptor is shown in black, and CXCL12 is colored differently in each model. The underlying numerical data for each figure panel can be found in S1 Data. NMR, nuclear magnetic resonance.
(TIF)

**S12 Fig. Alternative conformations of the CXCR4 N-terminus captured by the docking simulations.** Shown are representative conformations in which the distal N terminus of CXCR4 was found in proximity of the CXCL12 N-loop (A) in the context of the CXCL12 monomer or (B) in the context of the CXCL12 dimer. In panel (B), the distal CXCR4 N terminus potentially interacts with the N-loop of the CXCL12 dimer partner if fully extended. The receptor and CXCL12 are shown in black and purple, respectively. The second monomer in the CXCL12 dimer is shown in orange. Residue proximities reconciled by these alternative models but not by the best-scoring model are shown as light-blue dotted lines.
(TIF)

**S13 Fig. Benchmarking comparison of proposed and previously published CXCR4-CXCL12 complex geometries.** For each model, pairwise residue Cβ-Cβ distances were ranked and receiver operating characteristic curves were generated based on their ability to recognize: (A) the top 10% (14 crosslinks), (B) the top 25% (36), or (C) the top 50% (72) strongest experimentally determined cross-links. Model "Ngo" is our proposed model in this study (S2 Data), and "Tamamis" and "Ziarek" are models published previously [25,51]. The underlying numerical data for each figure panel can be found in S1 Data.
(TIF)

**S14 Fig. Surface expression levels of CXCR4 D20R and E26R mutants are comparable to WT CXCR4.** Surface expression was determined by flow cytometry using an allophycocyanin-conjugated anti-CXCR4 antibody and normalized to WT CXCR4. Data represent mean and SEM of $n$ = 4 independent replicates. The underlying numerical data for each figure panel can be found in S1 Data. WT, wild type.
(TIF)

**S15 Fig. The energy function of ICM local distance restraints.** The profile of the restraint penalty features a flat well of varying depth (defined by the restraint weight) at shorter interatomic distances and increases as these distances increase; however, rather than growing indefinitely, the penalty asymptotically approaches 0 as the interatomic distances continue to increase. This allows the sampling procedure to ignore those restraints that cannot be satisfied concurrently with the majority of other restraints and, in this way, resolve conflicts in the experimental data. The underlying numerical data for each figure panel can be found in S1 Data.
(TIF)

**S1 Table. Strategies for determining receptor-ligand complex geometry.**
(DOCX)

**S2 Table. Physicochemical properties of chemokine CXCL12 based on its crystal structures.**
(DOCX)

**S3 Table. Conformation clustering of CXCL12 crystal structures.**
(DOCX)

**S4 Table. Cross-linking efficiencies and distance restraint weights of receptor-chemokine cysteine pairs.**
(DOCX)

**S1 Data. Spreadsheet containing individual sheets of the underlying numerical data for panels in Figs 2, 3, 5 and S3, S4, S7, S11, S13, S14, S15 Figs.**
(XLSX)

**S2 Data. The coordinates of the CXCR4-CXCL12 complex presented here, in PDB format.**
(PDB)

## Acknowledgments

Authors would like to thank Drs. Catherina Salanga and Christopher Schafer (UC San Diego) for their help with recombinant chemokine purification, Dr. Åge Skjevik (University of Bergen) for helpful suggestions on model development, and the Trejo Lab (UC San Diego) for access to the TriStar multi-label plate reader.

## Author Contributions

**Conceptualization:** Tracy M. Handel, Irina Kufareva.

**Data curation:** Tony Ngo, Bryan S. Stephens, Tracy M. Handel, Irina Kufareva.

**Formal analysis:** Tony Ngo, Bryan S. Stephens, Irina Kufareva.

**Funding acquisition:** Tracy M. Handel, Irina Kufareva.

**Investigation:** Tony Ngo, Bryan S. Stephens, Irina Kufareva.

**Methodology:** Tony Ngo, Martin Gustavsson, Lauren G. Holden, Ruben Abagyan, Tracy M. Handel, Irina Kufareva.

**Project administration:** Tracy M. Handel, Irina Kufareva.

**Resources:** Ruben Abagyan, Tracy M. Handel, Irina Kufareva.

**Software:** Tony Ngo, Ruben Abagyan, Irina Kufareva.

**Supervision:** Tracy M. Handel, Irina Kufareva.

**Validation:** Tony Ngo, Bryan S. Stephens, Irina Kufareva.

**Visualization:** Tony Ngo, Irina Kufareva.

**Writing – original draft:** Tony Ngo, Bryan S. Stephens, Tracy M. Handel, Irina Kufareva.

**Writing – review & editing:** Tony Ngo, Martin Gustavsson, Tracy M. Handel, Irina Kufareva.

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
