## [Editor Report · Decision Letter 0]

18 Aug 2019

Dear Dr Kufareva, 

Thank you for submitting your manuscript entitled "Crosslinking-guided geometry of a complete CXC receptor-chemokine complex and the basis of chemokine subfamily selectivity" for consideration as a Research Article by PLOS Biology.

Your manuscript has now been evaluated by the PLOS Biology editorial staff, as well as by an academic editor with relevant expertise, and I'm writing to let you know that we would like to send your submission out for external peer review.

*Please be aware that, due to the voluntary nature of our reviewers and academic editors, manuscripts may be subject to delays during the holiday season. Thank you for your patience.*

Please re-submit your manuscript within two working days, i.e. by Aug 20 2019 11:59PM.

Kind regards,

Roli Roberts

Senior Editor

PLOS Biology

---

## [Decision Letter · Decision Letter 1]

17 Sep 2019

Dear Dr Kufareva,

Thank you very much for submitting your manuscript "Crosslinking-guided geometry of a complete CXC receptor-chemokine complex and the basis of chemokine subfamily selectivity" for consideration as a Research Article at PLOS Biology. Your manuscript has been evaluated by the PLOS Biology editors, an Academic Editor with relevant expertise, and by four independent reviewers.

In light of the reviews (below), we are pleased to offer you the opportunity to address the comments from the reviewers in a revised version that we anticipate should not take you very long. We will then assess your revised manuscript and your response to the reviewers' comments and we may consult the reviewers again. IMPORTANT: Several of the reviewers request that you improve the presentation of your paper, both in terms of the clarity and structure of the text, and helpfulness of the Figures; this is particularly important given the broad readership of our journal, and we are looking for significant improvement.

Your revisions should address the specific points made by each reviewer. Please submit a file detailing your responses to the editorial requests and a point-by-point response to all of the reviewers' comments that indicates the changes you have made to the manuscript. In addition to a clean copy of the manuscript, please upload a 'track-changes' version of your manuscript that specifies the edits made. This should be uploaded as a "Related" file type. You should also cite any additional relevant literature that has been published since the original submission and mention any additional citations in your response. 

Before you revise your manuscript, please review the following PLOS policy and formatting requirements checklist PDF: http://journals.plos.org/plosbiology/s/file?id=9411/plos-biology-formatting-checklist.pdf. It is helpful if you format your revision according to our requirements - should your paper subsequently be accepted, this will save time at the acceptance stage.

Please note that as a condition of publication PLOS' data policy (http://journals.plos.org/plosbiology/s/data-availability) requires that you make available all data used to draw the conclusions arrived at in your manuscript. If you have not already done so, you must include any data used in your manuscript either in appropriate repositories, within the body of the manuscript, or as supporting information (N.B. this includes any numerical values that were used to generate graphs, histograms etc.). For an example see here: http://www.plosbiology.org/article/info%3Adoi%2F10.1371%2Fjournal.pbio.1001908#s5.

For manuscripts submitted on or after 1st July 2019, we require the original, uncropped and minimally adjusted images supporting all blot and gel results reported in an article's figures or Supporting Information files. We will require these files before a manuscript can be accepted so please prepare them now, if you have not already uploaded them. Please carefully read our guidelines for how to prepare and upload this data: https://journals.plos.org/plosbiology/s/figures#loc-blot-and-gel-reporting-requirements.

Upon resubmission, the editors assess your revision and assuming the editors and Academic Editor feel that the revised manuscript remains appropriate for the journal, we may send the manuscript for re-review. We aim to consult the same Academic Editor and reviewers for revised manuscripts but may consult others if needed.

We expect to receive your revised manuscript within one month. Please email us (plosbiology@plos.org) to discuss this if you have any questions or concerns, or would like to request an extension. At this stage, your manuscript remains formally under active consideration at our journal; please notify us by email if you do not wish to submit a revision and instead wish to pursue publication elsewhere, so that we may end consideration of the manuscript at PLOS Biology.

When you are ready to submit a revised version of your manuscript, please go to https://www.editorialmanager.com/pbiology/ and log in as an Author. Click the link labelled 'Submissions Needing Revision' where you will find your submission record. 

Sincerely,

Roli Roberts

Senior Editor

PLOS Biology

REVIEWERS' COMMENTS:

Reviewer #1: 

This is a nice work that characterizes the interaction between the N-terminus of the chemokine receptor CXCR4 and the chemokine CXCL12 using pairwise crosslinking on the intact full-length receptor in live insect cells combined to molecular modeling. The biological question is highly relevant, as this receptor exerts crucial physiological roles, but there are no structures available for any receptor of CXC chemokines to help rational design of targeted drugs. In particular, the work reveals the topology of interactions between the distal N-terminus of the receptor and the chemokine, which was largely unknown, given that this segment is disordered in all structures of chemokine receptors solved so far and that NMR studies based of the investigation of isolated receptor segments lead to contradictory results. 

Experimentally, authors investigate the interaction between the receptor N-terminus and the core of the chemokine [P2G]CXCL12, which is an antagonist. They build a preliminary low resolution model of the chemokine-receptor complex to be refined and validated with disulfide trapping. Based on the model, authors select subsets of positions in the two interaction partner. These positions are substituted with Cysteine and combined pair-wise to assess the occurrence of crosslinking. In total, authors co-express in sf9 cells about 200 pairs of Cys-CXCR4 and Cys-[P2G]CXCL12 mutants, including positive and negative controls. Crosslinking occurred in position-specific patterns and most results could be easily rationalized. 

I have a concern about the application of flow cytometry to detect the occurrence of crosslinking. Under the assumption that [P2G]CXCL12 dissociates from the receptor when not covalently bound, authors quantify the fluorescent signal given by the cells treated with an antibody targeted to the ligand. This is a delicate step, which in my opinion deserves a better explanation. It is unclear when the ligand that is not covalently bound should dissociate. According to the experimental section, there is no washing step. How do the authors know that this antagonist dissociates so fast? Are there data for the koff of this ligand? Also, how do the authors take into account the likely different expression level of the different Cys-receptor mutants and Cys-chemokines in the quantification of the flow cytometry data? In this frame, giving crosslinking efficiency with an accuracy to the second decimal place (Supplementary Table 4) may not be appropriate.

Authors select a smaller subset of ligand-receptor combinations and validate flow cytometry data using western blot to detect the occurrence of crosslinking. Author state, page 10 lines 207-208, that WB data were in line with flow cytometry data. I have troubles in agreeing with this statement. In the modeling experiments, author apply constraints for pairs displaying >65% crosslinking efficiency (65% in cytometry or WB?). In Figure 3, the pair G3C-receptor/L29-ligand would be a hit according to flow cytometry, but not according to WB analysis. There are striking differences also in the pair M24/E15, as well as S9/H17. Importantly, these outliers are “red spots” in the flow cytometry data. Supplementary Fig. 4 is somewhat misleading, because the worst outlier S18/N22 is in fact not problematic, as it appears as a clear hit using both methods. 

It is not clear how author treated these discordant data when refining their model. For instance, was S9/H17 treated as a hit or not? According to the WB data, this pair is not a hit, so that data in figure 2f do not contradict the model. It may be worth revisiting page 12, line 272, but also page 13, lines 281 and following, taking into major account the WB data. 

It would instead be interesting to have an overview of the experimental data against the distribution of the ligand-receptor distances measured in the final model. In any case, I recommend to soften the statement at page 10, lines 218-220.

Finally, authors validate the model by mutating charged residues of the receptor predicted to come into proximity of reversely charged residues in the ligand. Did author perform also charge exchange experiments?

Overall, the paper is very interesting and relevant in the field. Molecular models based on crosslinking constraints are the best possible approximation when 3D structures are missing. Indeed, the model developed in this work reveals a number of interactions that were previously unknown and are nonetheless in line with previous functional studies. Therefore, the model is quite strong and it is very thoroughly discussed in the frame of existing literature in the discussion. Although the work is based on extensive and solid data, the handling of the data is in some places questionable, as discussed above. With respect to the scholar presentation, the manuscript indulges in a lot of details that make it difficult for the reader to gain a comprehensive overview of the story and make the overall manuscript very heavy. Moreover, there are parts in the experimental section that in fact belong to the discussion, for instance major sections of the paragraphs between lines 257-290. I am pretty confident that most of these issues can be resolved through a deep revision of the written text. This can become a really good paper.

Minor issues

Page 7, line 133. A preliminary sentence providing a first overview of the different blocks of mutants that have been tested would help the reader to go through the following section.

Page 7, line 135. Isn’t here 30? In Fig. 2A I count 42 when controls are included. 

Page 7, line 150-151. I find this sentence confusing, the aim is to say the occurrence and intensity of crosslinking was position-dependent, right? 

I could not find the information about the length of the distance restraints?

Reviewer #2:

In this manuscript, Ngo and coworkers use cysteine cross-linking to determine the geometry of the complex between CXCL12 and full-length CXCR4. It has been an open question in the field as to the mechanism that chemokine receptors use to bind their ligands. Historically it was thought that this binding occurred through distinct sites (CRS 1 and 2), although crystal structures then defined a CRS 1.5 site between CRS 1 and 2. In a subsequent study (Nat Commun 2017), this same group defined a CRS0.5 site between ACKR3 and CXCL12 through radiolytic footprinting, disulfide crosslinking and mutagenesis with the N-terminus of the receptor forming an antiparallel with the beta sheet of the CXCL12. They observe a similar structure here between CXCR4 and CXCL12. The approach used for the disulfide crosslinking appears to be significantly different, in this work using a flow cytometry-based assay, allowing a wide array of disulfide pairs to be tested. For these experiments, the antagonist [P2G]CXCL12 was used because CXCL12 does not form high affinity complexes with the receptor in the absence of G protein. The authors used an orthogonal assay (non-reducing SDS-PAGE) to confirm the crosslinking identified from flow cytometry, with very good agreement between the two approaches. They then performed modeling with those restraints to develop a model for the CXCR4:CXCL12 compex, identifying a number of important charge-charge interactions and asparagine interactions with sulfotyrosine. They then validated their predictions from the model in a functional assay by mutating residues that are predicted to form salt bridges – CXCR4 D20R and E26R. Both mutants displayed a right-shifted concentration response for arresetin recruitment and one with decrease in Emax.

Overall, these are important findings as the authors relate this model to differences in receptor binding to CXC and CC chemokines and also demonstrate the superiority of their model to previous attempts in the literature. From that perspective, it is an important addition to the literature. The previous model of ACKR3:CXCL12 does detract from its novelty, as the structure appears very similar in terms of the CRS0.5 structure.

Minor comment:

- Supplementary Table 4. Crosslinking efficiency is shown without any estimate of errors. That should be included as the data is clearly available (error bars are shown in Fig. 3c)

Reviewer #3:

The N-terminus of chemokine receptors is known to play an important role in its affinity and selectivity for chemokines. The authors studied the interaction between the chemokine receptor CXCR4 and the chemokine CXCL12, as one of the best characterised receptor-chemokine model system, using a combination of molecular modelling and cysteine cross-linking. The resulting model rotates the chemokine relative to the receptor relative to the previous models, and that allows in buried in the TM bumble N terminus of the chemokine to occupy additional binding pocket. This provides a potential explanation of the of CXC receptor-chemokine selectivity against other subfamilies.

Overall, this is an interesting paper and I sure it will catalyse further structural bioinformatics work to expand these mode of binding to other chemokine-chemokine receptor pairs. 

The following comments may improve the manuscript so that it would be easier to read and therefore would have a larger impact for the intended audience: 

1) Overall, it is rather difficult to read. This could easily be improved by another round of editing with a view of the international audience that is the likely audience. 

2) The computational model on which authors base their selection of cross-linking pairs, is not very well presented in this manuscript. There is only a global view presented on Fig 1, and the details are missing. The Fig 1 of the accompanying manuscript is , on the other hand, a much better description of this model. 

3) P16 l335-336 .. “we mapped, in a comprehensive and unbiased manner, pairwise residues…”. This should be discussed in depth as this type of mapping may not not as unbiased as one would think. It is based on a model that the authors have built initially to restrict the number of potential cross-linking pairs to be studied. Unfortunately, cross-linking is prone to stabilise some transient interactions. It is, effectively, searching for the lost keys under the lamppost, and may introduce a model bias through the “streetlight effect”. It would be important to highlight the cross-linking pairs or their absence that do confirm that the cross-linking is specific and is not influenced by the model. This, for example, could be argued by showing the data for a significant number of negative controls (comparable to the number of positive cross-links). How different was the final experimental-results constrained model to the initial prediction? 

4) As is clear from figure 3b/c and S4 that there are some significant disagreements between the flow cytometry and the western blot data. What is the correlation coefficient between the data? Could the authors explain what are the reasons these differences? Could the authors elaborate why one experiment would provide better accuracy than the other? The authors seem to assume the western blot data is more reliable: what is the agreement between the final model and the western blot cross linking data (eg, fig S9 and S11?).

Minor issues:

1) The authors claim in the abstract to have found the basis for chemokine selectivity that is cause by an alternative positioning of the N -terminus of the chemokine in the TM bundle of the receptor. It is worth explaining in the abstract how this is determined by the binding mode of the N-terminus of the receptor (eg, rotational positioning of the chemokine molecule relative to the receptor). 

2) p5-6 Several characteristics of the CXCL12 model are listed. The readability would greatly improve if the relevant subfigures of figure 1 are mentioned for each bullet point. 

3) On p8, line 159-160, the authors reference figure 1, but it is not clear from figure 1 what “the proposed structural role of L26 and I28” is.

4) P10 l224 Wrong reference: Fig S1 does not contain a model of the CXCR4-CXCL12 complex. 

5) P24 l547: The formula contains the word “ratio”. This might be confusing, since the ratio has already been written out (by the “/” character). What software was used to analyse the Western blots and the SDS-page densitometry? What bands were taken into account for the densitometry reading? How does this give more information that IR800/IR680?

6) P25 l563: The authors use “weighted local harmonic distance restraints” for their model. Is this based on both the flow cytometry data and/or the western blot data? If both, how were they combined?

7) Fig. 2: Please add explanation of the grey squares in the heatmap. Intuitively, one would interpret is a “normal” or 100% of the reference, but 100% is shown in red.

Reviewer #4:

The manuscript by Ngo et al. takes a very thorough and robust experimental approach to delineate the geometry of CXC receptor-chemokine complexes. Such assemblies have emerged as major therapeutic targets, however, structural models of complete assemblies have been lacking

.

As such, this study is to my knowledge the first to attempt modeling of such complete receptor-chemokine assemblies based on experimental data. 

Importantly, the study by Ngo et al. has led to novel findings which are set to reshape the way the field is interrogating receptor-chemokine complexes. The study does have some overlap with the accompanying manuscript but is important and novel enough in its own right.

(1) The study would benefit from demonstrating that for the top cross-linking cysteine couples, computational models can be obtained wherein the two residues are mutated to cysteines which approach each other within disulfide-bonding distances. Such structural analysis is important to provide a true structural context for the cross-linking approach.

(2) The readability of the manuscript could benefit from a structural model, preferably in schematic mode, that summarizes the current understanding of CXC receptor-chemokine assemblies in terms of the site 1/site 2 models. 

Additional comments to improve display items:

-The electrostatic potential scale used in Fig. 1g for CXCL12 basically suggests that the entire face shown is covered with positively charged residues. This cannot be correct. The authors will need to specify the electrostatic potential scale shown and how this calculation was done.

-Supplementary Table 2 is lacking units of Å2 for the B-factors.

-Supplementary Table 3 is lacking units of Å for the resolution column.

-Several of the figures feature color combinations that are difficult to discern. For instance, Supplementary Fig. 6 combines purple with black for the chemokine and receptor, respectively.

---

## [Decision Letter · Decision Letter 2]

20 Jan 2020

Dear Dr Kufareva,

Thank you for submitting your revised Research Article entitled "Crosslinking-guided geometry of a complete CXC receptor-chemokine complex and the basis of chemokine subfamily selectivity" for publication in PLOS Biology. I have now obtained advice from two of the original reviewers and have discussed their comments with the Academic Editor. 

We're delighted to let you know that we're now editorially satisfied with your manuscript. However before we can formally accept your paper and consider it "in press", we also need to ensure that your article conforms to our guidelines. A member of our team will be in touch shortly with a set of requests. As we can't proceed until these requirements are met, your swift response will help prevent delays to publication. Please also make sure to address the Data and other policy-related requests noted at the end of this email.

*Copyediting*

*Published Peer Review History*

*Early Version*

*Submitting Your Revision*

Sincerely,

Roli Roberts

Senior Editor

PLOS Biology

DATA POLICY:

Regardless of the method selected, please ensure that you provide the individual numerical values that underlie the summary data displayed in the following figure panels as they are essential for readers to assess your analysis and to reproduce it: Figs 2ACE, 3BC, 5BDF, S3, S4, S6A, S11, S13, S14, S15. We note that the existing file S2_Data contains data for Figs 2, 3, 5, S3 and S14. However, we can't see data underlying Figs S4, S6A, S11, S13 and S15; please could you supply this, or clarify where it can be found?

REVIEWERS' COMMENTS:

Reviewer #1:

The Authors have addressed most of the concerns expressed in the first revision and thoroughly revised the work. The current manuscript meets the standards for publication in PLOS Biology. Nice paper. 

Reviewer #3:

[identifies himself as Dmitry B. Veprintsev]

This is much clarified version of the manuscript that addresses my comments.

---

## [Editor Report · Decision Letter 3]

2 Mar 2020

Dear Dr Kufareva,

On behalf of my colleagues and the Academic Editor, Mathieu JM Bertrand, I am pleased to inform you that we will be delighted to publish your Research Article in PLOS Biology. 

Early Version

PRESS 

Kind regards,

Alice Musson

Publication Assistant, 

PLOS Biology

on behalf of

Roland Roberts,

Senior Editor

PLOS Biology